# TOWARD NEURAL STREAMING SCHEDULING: A MEMORY-AUGMENTED REINFORCEMENT LEARNING MODEL WITH CRITICAL STRUCTURE ENCODING

## ABSTRACT

Many large-scale data analytics and AI systems execute jobs structured as Directed Acyclic Graphs (DAGs), which encode precedence constraints among interdependent stages. Efficient DAG scheduling is crucial for maximizing system throughput, especially in streaming settings where diverse jobs arrive continuously and require real-time decisions. Despite progress by heuristic and learning-based scheduling methods, capturing execution-critical structures and leveraging historical scheduling context remain key challenges. Building on this motivation, we propose MACE, a Memory-Augmented reinforcement learning model with Critical structure Encoding, which implements a scheduling policy for streaming jobs by sequentially selecting runnable stages and assigning parallelism based on cluster state. The policy is trained to minimize average job completion time using defined rewards. Specifically, MACE consists of two core components: (i) CSformer builds hierarchical embeddings that integrate stage-job-global information, capturing execution-critical structures through critical-path-aware positional encodings and an extended attention field. This design guides the policy toward latency-sensitive and structurally related stages. (ii) A memory-augmented scheduler then uses the learned embeddings and a job memory to exploit historical contexts for the final stage and parallelism selection. Extensive experiments on Spark using the TPC-H benchmark demonstrate that MACE outperforms state-of-the-art baselines by up to 9.38% under diverse workload conditions.

## 1 INTRODUCTION

The explosive growth of large-scale data analytics and Artificial Intelligence (AI) workloads has spurred the evolution of distributed processing systems such as Spark (Zaharia et al., 2012) and Flink (Carbone et al., 2015). In these systems, cluster schedulers play a central role in coordinating execution and optimizing efficiency (Grandl et al., 2014), and their effectiveness becomes increasingly critical as clusters and job volumes scale (Verma et al., 2015; Moritz et al., 2018).

Many computational workloads are structured as *Directed Acyclic Graphs (DAGs)*, where nodes denote processing stages and edges encode precedence constraints (Topcuoglu et al., 2002; Sinnen, 2007). The goal of scheduling is to assign stages to a limited pool of identical executors to satisfy dependencies and minimize average Job Completion Time (JCT). *Static scheduling* (Kwok & Ahmad, 1999; Jeon et al., 2023; Shahout & Mitzenmacher, 2024) assumes all DAGs are known upfront and plans execution ahead of time. Various heuristic methods have been proposed for this setting, such as fair scheduling (Isard et al., 2009) and list scheduling (Topcuoglu et al., 2002). Relying on fixed rules or cost metrics, these approaches often overlook the structural complexity of DAGs and lack adaptability to dynamic environments.

A more practical setting is *streaming scheduling*, where DAGs arrive continuously with varying structures and job characteristics, requiring real-time decisions under dynamic workloads and limited resources. To tackle this, learning-based methods have gained attention by framing scheduling as a reinforcement learning problem. As a representative work, Decima (Mao et al., 2019) employs Graph Neural Networks (GNNs) (Kipf & Welling, 2017; Veličković et al., 2018) to encode cluster states, and learns a scheduling policy via interaction with the environment and received rewards.

Follow-up studies extend this framework to heterogeneous resources (Lin et al., 2022; Wang et al., 2025) and incorporate techniques like task duplication (Zhou et al., 2022).

Despite remarkable progress, existing methods neglect two key factors for efficient scheduling. (i) Lack of explicit modeling for *execution-critical structures*, such as critical paths and long-range dependencies. Stages on the critical path determine the earliest possible JCT (Kelley Jr & Walker, 1959), and long-range dependencies affect when stages can run and how resources are shared (Anantharam, 1999). Recent methods (Gagrani et al., 2022; Luo et al., 2023) either omit them or only consider direct children (Xu et al., 2018; Wu et al., 2020), rendering them ill-suited for complex scheduling cues. (ii) Absence of a mechanism to incorporate *historical scheduling contexts*. Real-world workloads often exhibit recurring patterns (Liu et al., 1987), and past scheduling decisions offer valuable context for future ones (Sutton, 1984; Sutton et al., 1998). Yet, neural models trained via gradient descent are inherently short-sighted (Kirkpatrick et al., 2017; Kemker et al., 2018), limiting their ability to retain long-term scheduling trajectories.

To fill this gap, we propose **MACE**, a *Memory-Augmented reinforcement learning model with Critical structure Encoding* for streaming scheduling. MACE adopts a reinforcement learning framework to learn a scheduling policy that sequentially selects runnable stages and assigns parallelism (number of executors) based on the cluster state. The policy is trained end-to-end via policy gradient to minimize average JCT, using rewards derived from observed job completions in the Spark environment. Specifically, MACE consists of two core components: (i) *CSformer* encodes the cluster state into hierarchical embeddings capturing stage-job-global information. It emphasizes execution-critical structures via critical-path-aware positional encodings and an extended attention field, helping the policy prioritize latency-sensitive stages. (ii) A *memory-augmented scheduler* selects stages and parallelism by jointly attending to the learned embeddings and a compact job memory, which stores historical scheduling decisions and outcomes to support more context-aware decision-making.

In summary, our contributions are three-fold:

- We propose a novel reinforcement learning model MACE for streaming scheduling. It incorporates execution-critical structures and historical scheduling contexts, allowing the scheduler to account for structural priorities and prior decisions in dynamic cluster environments.

- We design two key components within MACE: (i) CSformer, which captures execution-critical structures using critical-path-aware positional encodings and an extended attention field; and (ii) a memory-augmented scheduler, which integrates a compact job memory with learned embeddings to guide decision-making based on recent scheduling decisions and outcomes.

- Extensive experiments on Spark with the TPC-H benchmark demonstrate that MACE reduces average job completion time by up to 9.38% over state-of-the-art baselines across diverse workloads.

## 2 RELATED WORK

Scheduling methods can be broadly grouped into *streaming scheduling* and *static scheduling* (Singh et al., 2015). In streaming scheduling, jobs arrive continuously and future arrivals are unknown. The scheduler observes the current cluster state and runnable jobs, and makes online decisions about which stage to run next and how much parallelism to allocate. In contrast, static scheduling (Gu et al., 2024; Qi et al., 2025; Xu et al., 2025) assumes a known set of DAGs provided in advance and generates a complete task ordering before execution begins. Such methods are not designed to handle streaming arrivals or predict parallelism dynamically, and therefore cannot be directly applied to the streaming setting. Here we focus on the more practical and challenging case of streaming scheduling and review prior work in this area.

**Heuristic Scheduling**   Heuristic algorithms have long underpinned streaming scheduling, generating execution plans from manually crafted rules and cost models. They are typically categorized into list-based (Bittencourt et al., 2010; Arabnejad & Barbosa, 2013; AlEbrahim & Ahmad, 2017), cluster-based (Yang & Gerasoulis, 1991; Bajaj & Agrawal, 2004), and task duplication-based (Ranaweera & Agrawal, 2000; Shin et al., 2008) strategies. List-based methods rank tasks by static or dynamic metrics and assign them via heuristics such as the earliest finish time. Notable examples include HEFT (Topcuoglu et al., 2002), which uses upward rank to estimate task criticality, and DLS (Sih & Lee, 2002), which selects task-processor pairs based on level-based scoring. Cluster-based methods (Kim, 1988; Park et al., 1997) group interdependent tasks to reduce

communication but risk load imbalance when clusters misalign with resource availability. Task duplication-based methods (Darbha & Agrawal, 2002; He et al., 2018) replicate parent tasks across nodes to mitigate data transfer delays. While easy to implement and interpretable, these heuristics rely on fixed rules, making them inflexible in dynamic environments.

**Learning-based Scheduling**  The advent of deep learning (LeCun et al., 2015; Vaswani et al., 2017) has spurred growing interest in learning-based scheduling beyond static heuristics. Among them, deep reinforcement learning (Mnih et al., 2015; Lillicrap et al., 2015) has emerged as a dominant paradigm, framing scheduling as a Markov Decision Process (MDP) optimized through interaction with the environment. A seminal work is Decima (Mao et al., 2019), which leverages graph neural networks to encode DAG structure and learn sequential scheduling decisions. Building on this foundation, LACHESIS (Zhou et al., 2022) incorporates task duplication heuristics, DREAM (Ni et al., 2020) learns throughput-aware embeddings for dynamic placement, and DeepWeave (Sun et al., 2021) improves coflow-level scheduling via transmission modeling. These approaches offer improved adaptability and generalization over heuristics, particularly in dynamic environments. However, they often struggle to capture execution-critical structures and long-range dependencies, both of which are crucial for efficient scheduling.

## 3  PROBLEM DESCRIPTION

**Streaming Scheduling**  Modern data processing systems (Zaharia et al., 2012; Carbone et al., 2015; Rocklin et al., 2015) like Spark and Flink represent complex jobs as DAGs of computation stages and dependencies. These systems operate in a streaming mode, where jobs are submitted at irregular intervals over time, as illustrated in Figure 1, and require real-time scheduling over a shared resource pool.

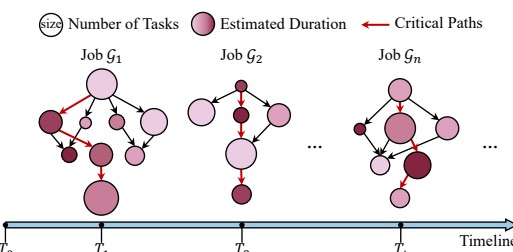

Figure 1: Job DAGs arrive over time at irregular intervals. Each node represents a stage with task count and duration, while red paths highlight *critical paths* that dominate JCT.

Formally, a job stream is denoted as $\mathcal{J} = \{J_1, J_2, \ldots\}$, where job $J_i$ is defined by a DAG $\mathcal{G}_i = (\mathcal{V}_i, \mathcal{E}_i)$. A stage $v \in \mathcal{V}_i$ has processing time $p_v$, and an edge $(u, v) \in \mathcal{E}_i$ imposes a precedence constraint that $u$ must finish before $v$ starts. Jobs are executed on a cluster of $m$ identical executors $\mathcal{M} = \{M_1, \ldots, M_m\}$, where stage execution is exclusive and non-preemptive.

At each scheduling decision step, the scheduler must (i) select a runnable stage to execute from active jobs, and (ii) allocate a number of executors to determine its parallelism. A key factor in these decisions is the *critical path* of job DAGs, defined as the longest path in terms of cumulative processing time $\text{CP}(\mathcal{G}_i) = \max_P \sum_{v \in P} p_v$, where $P = \{v_1, \ldots, v_k\}$ is a valid path such that $(v_i, v_{i+1}) \in \mathcal{E}_i$. As the critical path (Kelley Jr & Walker, 1959) sets the earliest possible completion time of a job, emphasizing its stages is crucial to achieve low-latency execution. Let $s_v$ and $c_v = s_v + p_v$ be the start and completion timestamps of stage $v$, and define the completion time of job $\mathcal{G}_i$ as $C_i = \max_{v \in \mathcal{V}_i} c_v$. The goal is to learn a scheduling policy $\pi$ that respects precedence constraints and resource limits while minimizing average JCT across all jobs:

$$\min_{\pi} \quad \frac{1}{|\mathcal{J}|} \sum_{J_i \in \mathcal{J}} C_i(\pi). \tag{1}$$

**Graph Features**  To support effective policy learning, hierarchical features are extracted for each DAG $\mathcal{G}_i = (\mathcal{V}_i, \mathcal{E}_i)$. At the *job level*, three features summarize the global scheduling state of job $\mathcal{G}_i$: (i) resource occupancy: the number of executors assigned to $\mathcal{G}_i$; (ii) priority indicator: a binary flag indicating whether $\mathcal{G}_i$ is currently selected by the scheduler; (iii) resource availability: the number of idle executors in the cluster. At the *stage level*, each stage $v \in \mathcal{V}_i$ inherits job-level features from its corresponding job and appends two stage-specific features: (i) remaining workload: the total estimated duration of unexecuted tasks of $v$; (ii) task progress: the number of unexecuted tasks.

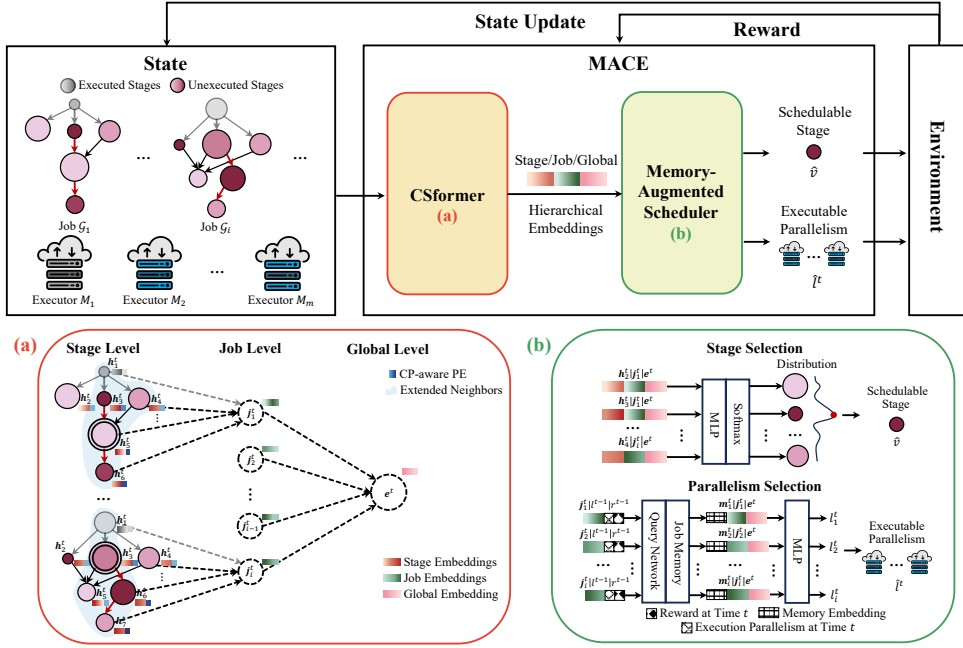

Figure 2: **Overview of MACE's architecture**. MACE takes the cluster state as input, including job DAGs and executor availability, and learns a policy to schedule streaming jobs. It first uses (a) *CSformer* to extract hierarchical embeddings at stage-job-global levels, where execution-critical structures are emphasized through critical-path-aware positional encodings and an extended attention field. The (b) *memory-augmented scheduler* fuses the learned embeddings with job memory to select a schedulable stage and its executable parallelism. The environment updates the cluster state based on these scheduling decisions and returns rewards, which are used to train the policy end-to-end via policy gradient to minimize average JCT.

# 4 MACE METHODOLOGY

This section presents the novel MACE model designed for streaming scheduling. It begins with an architectural overview, and then details CSformer capturing execution-critical structures. The memory-augmented scheduler is introduced next, followed by training procedures.

## 4.1 OVERVIEW

MACE formulates streaming scheduling as a *Reinforcement Learning (RL)* problem as illustrated in Figure 2, where scheduling decision steps are triggered by events such as job arrivals or stage completions. At each step, the cluster state, including job DAGs and executor availability, is represented as graphs with topology and associated features (see Section 3), serving as input to the model.

To encode the cluster state, *CSformer* learns hierarchical embeddings at stage-job-global levels. Specifically, to emphasize execution-critical structures, it computes critical-path-aware positional encodings based on the depth of each stage along the critical path, and defines an extended attention field using shortest-path distances to capture long-range dependencies. These structural cues are aggregated through multi-layer attention to obtain stage-level embeddings. To support cross-level information exchange, virtual job and global nodes are added as parent nodes, from which job-level and global-level embeddings are derived via aggregation from their respective children. Based on these learned embeddings, the *memory-augmented scheduler* scores candidate stages and selects one for execution. It then determines the executable parallelism for the selected stage's job by retrieving job memory using the job-level embedding and recent reward signals, where the memory stores historical scheduling decisions and outcomes to provide temporal context.

The environment updates the cluster state according to scheduling decisions and computes rewards based on job completion times, repeating this process until all runnable stages are scheduled or all executors are assigned. The entire model is trained end-to-end using policy gradient. Without loss of generality, all subsections below describe the model at scheduling decision step $t$.

## 4.2 CSFORMER

Based on the graph topologies and features derived from the current cluster state, CSformer builds hierarchical embeddings at stage-job-global levels. These embeddings capture execution-critical structures, supporting downstream scheduling with structural and contextual awareness.

**Stage Level**   Each stage $v$ is initialized with a feature vector $\mathbf{x}_v^t \in \mathbb{R}^5$, encoding runtime attributes as defined in Section 3. To capture execution-critical structures, we first compute critical-path-aware positional encodings and define the extended receptive field. Particularly, each stage is assigned a positional embedding based on its critical-path depth $\mathrm{CP}_d(v)$, representing the distance from the DAG's entry point along the critical path (set to 0 for non-critical-path stages). Following the original Transformer formulation (Vaswani et al., 2017), the positional encoding is computed as:

$$PE_{(v,2i)} = \sin\left(\frac{\mathrm{CP}_d(v)}{10000^{2i/d}}\right), \quad PE_{(v,2i+1)} = \cos\left(\frac{\mathrm{CP}_d(v)}{10000^{2i/d}}\right), \tag{2}$$

where $d$ is the embedding dimension and $i$ indexes the dimensions. Each stage attends to an extended receptive field defined by:

$$\mathcal{N}_k^{\mathrm{unexec}}(v) = \{u \in \mathcal{V} : \mathrm{dist}(v, u) \leq k \text{ and } u \text{ is unexecuted}\}, \tag{3}$$

where $\mathrm{dist}(v, u)$ is the shortest path between $v$ and $u$, computed without regard to edge direction. This design captures multi-hop parent-child relations that shape scheduling priorities while restricting attention to active neighbors.

In each layer of the CSformer encoder stack, the model performs the following multi-head attention. For each head $h$, we compute the query, key, and value vectors as:

$$\mathbf{q}_v^{(h)} = W_Q^{(h)}(\mathbf{x}_v^t + PE_v), \quad \mathbf{k}_u^{(h)} = W_K^{(h)}(\mathbf{x}_u^t + PE_u), \quad \mathbf{v}_u^{(h)} = W_V^{(h)}(\mathbf{x}_u^t + PE_u), \tag{4}$$

where $\{W_Q^{(h)}, W_K^{(h)}, W_V^{(h)}\} \in \mathbb{R}^{d \times d_h}$ are learnable projection matrices. The attention weights and output for head $h$ are:

$$\alpha_{vu}^{(h)} = \frac{\exp\left(\langle \mathbf{q}_v^{(h)}, \mathbf{k}_u^{(h)} \rangle / \sqrt{d_h}\right)}{\sum\limits_{w \in \mathcal{N}_k^{\mathrm{unexec}}(v)} \exp\left(\langle \mathbf{q}_v^{(h)}, \mathbf{k}_w^{(h)} \rangle / \sqrt{d_h}\right)}, \quad \mathbf{z}_v^{(h)} = \sum_{u \in \mathcal{N}_k^{\mathrm{unexec}}(v)} \alpha_{vu}^{(h)} \cdot \mathbf{v}_u^{(h)}, \tag{5}$$

Outputs from all $H$ heads are concatenated and projected to form the stage-level embedding:

$$\mathbf{h}_v^t = W_O \left[\mathbf{z}_v^{(1)} \| \cdots \| \mathbf{z}_v^{(H)}\right], \tag{6}$$

where $W_O \in \mathbb{R}^{d \times (H \cdot d_h)}$ is the output projection matrix.

**Job and Global Level**   To obtain job- and global-level embeddings, virtual nodes are added on top of stage-level embeddings: each job node connects to all stages $v \in \mathcal{V}_i$ in its DAG $\mathcal{G}_i$, and all job nodes connect to a shared global node.

Each job node is initialized with a raw feature vector $\mathbf{x}_{j_i}^t \in \mathbb{R}^3$, encoding resource-related attributes and cluster-wide availability (Section 3). Its embedding $\mathbf{j}_i^t \in \mathbb{R}^d$ is computed by aggregating stage-level embeddings $\mathbf{h}_v^t$ from the unexecuted stages within the corresponding DAG:

$$\mathbf{j}_i^t = g_j \left(\sum_{v \in \mathcal{V}_i^{\mathrm{unexec}}} f_j(\mathbf{h}_v^t)\right) + \mathbf{x}_{j_i}^t, \tag{7}$$

where $\mathcal{V}_i^{\mathrm{unexec}} \subseteq \mathcal{V}_i$ denotes the unexecuted stages in job $\mathcal{G}_i$, and $f_j(\cdot)$ and $g_j(\cdot)$ are learnable non-linear transformations. The global-level embedding $\mathbf{e}^t \in \mathbb{R}^d$ summarizes all job-level embeddings:

$$\mathbf{e}^t = g_e \left(\sum_i f_e(\mathbf{j}_i^t)\right), \tag{8}$$

with $f_e(\cdot)$ and $g_e(\cdot)$ as transformation functions. This hierarchical design enables CSformer to abstract stage-level dynamics into job- and global-level embeddings, supporting informed decisions.

## 4.3 MEMORY-AUGMENTED SCHEDULER

The scheduler leverages the learned stage-job-global embeddings and a compact job memory to makes two decisions: (i) selecting a runnable stage and (ii) determining execution parallelism for that stage's job. The memory stores recent scheduling trajectories, helping the scheduler incorporate historical contexts into current decisions.

**Stage Selection**   Based on the stage-job-global hierarchical embeddings, the scheduler selects a stage for execution by scoring stages in the runnable set $\mathcal{S}_t$, where a stage is *runnable* if all its parent stages have completed execution within the current job DAGs.

For each stage $v \in \mathcal{S}_t$ belonging to job $\mathcal{G}_i$, the priority score $s_v^t$ is computed via a learnable function $s(\cdot)$ over stage-level embedding $\mathbf{h}_v^t$, job-level embedding $\mathbf{j}_i^t$, and global-level embedding $\mathbf{e}^t$:

$$s_v^t = s\left(\left[\mathbf{h}_v^t \,\|\, \mathbf{j}_i^t \,\|\, \mathbf{e}^t\right]\right), \tag{9}$$

where $s(\cdot)$ is a multi-layer perceptron that maps the concatenated embedding to a scalar. The selection probability of stage $v$ is computed via softmax over all runnable stages:

$$P(v \mid \mathcal{S}_t) = \frac{\exp(s_v^t)}{\sum\limits_{u \in \mathcal{S}_t} \exp(s_u^t)}. \tag{10}$$

**Parallelism Selection**   To determine the execution parallelism, the scheduler retrieves job memory using the job-level embedding and the last reward signal, where the job memory stores historical scheduling decisions and outcomes to provide temporal context for adaptive decision-making.

The action space for job-level execution parallelism is defined as $\mathcal{L} = \{1, 2, \ldots, L\}$, where $L$ is the total number of available executors. For each job $\mathcal{G}_i$, the scheduler determines its parallelism $\ell_i^t \in \mathcal{L}$ by attending to a shared memory $\mathbf{M}_{t-1} \in \mathbb{R}^{d_m \times d_m}$, which is maintained as a fixed-size recurrent state summarizing all past decision trajectories up to step $t - 1$. This allows the memory to retain temporal experience and form a compact representation of how prior decisions influenced observed job dynamics. Specifically, to retrieve relevant memory, the scheduler first forms a query vector using its current embedding $\mathbf{j}_i^t$, previously selected parallelism $\hat{\ell}^{t-1}$, and the resulting reward $r^{t-1}$:

$$\mathbf{o}_i^t = [\mathbf{j}_i^t \,\|\, \hat{\ell}^{t-1} \,\|\, r^{t-1}]. \tag{11}$$

This query is passed through a learnable query network $q(\cdot)$ to attend over the memory matrix and obtain a memory-informed embedding:

$$\mathbf{m}_i^t = \mathbf{M}_{t-1} \cdot q(\mathbf{o}_i^t). \tag{12}$$

For each candidate parallelism $\ell \in \mathcal{L}$, a score is computed via a learnable function $\delta(\cdot)$ that consumes both memory and current context:

$$\delta_i^t(\ell) = \delta([\mathbf{m}_i^t \,\|\, \mathbf{j}_i^t \,\|\, \mathbf{e}^t], \ell). \tag{13}$$

The final parallelism decision is sampled from a softmax distribution over the score space:

$$P(\ell \mid \mathbf{o}_i^t) = \frac{\exp(\delta_i^t(\ell))}{\sum\limits_{\ell' \in \mathcal{L}} \exp(\delta_i^t(\ell'))}. \tag{14}$$

We compute the softmax only over parallelisms admissible for job $\mathcal{G}_i$. The criterion for admissibility and the projection to an executable allocation are detailed in Appendix A.

After selecting the next stage to execute, the scheduler updates the job memory using the context triplet $\hat{\mathbf{o}}^t = (\hat{\mathbf{j}}^t, \hat{\ell}^{t-1}, r^{t-1})$ from its associated job. Following prior work on memory writing (Katharopoulos et al., 2020; Morad et al., 2023; Le et al., 2024), the memory is updated by:

$$\mathbf{M}_t = \mathbf{M}_{t-1} \odot \mathbf{C}_\theta(\hat{\mathbf{o}}^t) + \mathbf{U}_\phi(\hat{\mathbf{o}}^t), \tag{15}$$

where $\odot$ denotes element-wise Hadamard multiplication. The calibration matrix $\mathbf{C}_\theta(\hat{\mathbf{o}}^t)$ modulates the retention of past memory, while the update matrix $\mathbf{U}_\phi(\hat{\mathbf{o}}^t)$ introduces new information based on the current observation. These two matrices are computed as:

$$\mathbf{C}_\theta(\hat{\mathbf{o}}^t) = 1 + \tanh(\boldsymbol{\theta}_t \otimes v_c(\hat{\mathbf{o}}^t)), \quad \mathbf{U}_\phi(\hat{\mathbf{o}}^t) = \eta_\phi(\hat{\mathbf{o}}^t)\left[v_u(\hat{\mathbf{o}}^t) \otimes k_u(\hat{\mathbf{o}}^t)\right]. \tag{16}$$

Here, $\{v_c, v_u, k_u\}$ are learnable projections mapping the context vector into memory space. The vector $\boldsymbol{\theta}_t \in \mathbb{R}^{d_\theta}$ is randomly sampled from a trainable matrix $\boldsymbol{\theta} \in \mathbb{R}^{128 \times d_\theta}$ for stability. The function $\eta_\phi$ outputs a scalar update gate, and $\otimes$ denotes the outer product.

### 4.4 MODEL TRAINING

Training proceeds over multiple *episodes*, each consisting of a sequence of scheduling decision steps. At each step $t \in \{1, \ldots, n\}$, where $n$ is the total number of steps in the episode, the policy probability $\pi(t)$ represents the likelihood of selecting the executed scheduling decision given the current cluster state. After executor allocation, a scalar reward $r^t$ is received, based on the average JCT (Chhajed & Lowe, 2008). Specifically, let $T^t$ be the wall-clock time at step $t$, and $J^t$ the number of jobs in the cluster during interval $[T^{t-1}, T^t)$. The reward is given by:

$$r^t = -(T^t - T^{t-1})J^t. \tag{17}$$

All modules in MACE are differentiable and jointly parameterized by $\Theta$. To optimize these parameters, we adopt the REINFORCE algorithm (Williams, 1992):

$$\Theta \leftarrow \Theta + \gamma \sum_{t=1}^{n} \nabla_\Theta \log \pi_\Theta(t) \left( \sum_{t'=t}^{n} r^{t'} - b^t \right), \tag{18}$$

where $\gamma$ is the learning rate, and $b^t$ is a baseline to reduce variance in gradient estimation (Weaver & Tao, 2001). A common choice sets $b^t$ as the cumulative reward from step $t$ onward (Greensmith et al., 2004), averaged across episodes. To ensure robustness under stochastic job arrivals, we gradually increase the episode length during training and terminate each episode at a random step sampled from an exponential distribution. This strategy discourages pathological behaviors and promotes the learning of generalizable scheduling policies.

## 5 EXPERIMENTS

**Datasets**   We leverage the TPC-H benchmark[1] (Mao et al., 2019), a gold-standard dataset for Spark stream-processing scheduling. Each DAG encodes the stage dependency graph for a job, with per-stage task counts and task-duration distributions estimated from empirical runs. We consider seven data scales $\{2g, 5g, 10g, 20g, 50g, 80g, 100g\}$, where g denotes gigabytes of raw data, and 22 query templates. Training episodes are generated online: each starts with $N_{\text{init}}^{\text{train}}$ pre-arrived jobs, followed by $N_{\text{stream}}^{\text{train}}$ jobs arriving via a Poisson process with mean inter-arrival $\Delta^{\text{train}}$. For each arrival, we uniformly sample a scale-template pair. The executor budget during training is fixed at $L^{\text{train}}=50$, with defaults $N_{\text{init}}^{\text{train}}=10$, $N_{\text{stream}}^{\text{train}}=100$, and $\Delta^{\text{train}}=25$ s. Test episodes adopt the same DAG generation process and vary the four-tuple $(N_{\text{init}}^{\text{test}}, N_{\text{stream}}^{\text{test}}, \Delta^{\text{test}}, L^{\text{test}})$ to evaluate performance under diverse cluster capacities and workload intensities.

**Evaluation Protocol**   We compare MACE with the following baselines: FIFO[2], SJF-CP (Shortest Job First with a Critical-Path proxy) (Kwok & Ahmad, 1999), WFS (naive Weighted Fair Scheduling)[3], HEFT (Heterogeneous Earliest Finish Time) (Topcuoglu et al., 2002), DRF (Dominant Resource Fairness) (Ghodsi et al., 2011), SRPT (Shortest Remaining Processing Time) (Harchol-Balter, 2013), and Decima (Mao et al., 2019). Our evaluation reports *average job completion time*, a universally accepted metric for scheduling efficiency in systems and queueing theory (Harchol-Balter, 2013). For details of experiments, including implementation and hyperparameters, refer to Appendix B. Note that learning-based schedulers such as LACHESIS (Zhou et al., 2022) and Deep-Weave (Sun et al., 2021) target heterogeneous task placement or coflow-level network scheduling under different system assumptions and objectives, so we treat them as complementary related work rather than direct baselines for our streaming cluster setting.

### 5.1 MAIN RESULTS AND ANALYSIS

**Robustness under Diverse Streaming Conditions**   Table 1 reports the average JCT across four stress-testing scenarios, each varying a single parameter while keeping others fixed. Specifically, we test: (i) *backlog robustness* by varying the number of pre-arrived jobs $N_{\text{init}}^{\text{test}} \in \{0, 30\}$ to examine

---

[1]https://www.tpc.org/tpch/

[2]https://spark.apache.org/docs/latest/job-scheduling.html

[3]https://hadoop.apache.org/docs/current/hadoop-yarn/hadoop-yarn-site/FairScheduler.html

Table 1: Quantitative results (avg. JCT $\pm$ standard deviation in seconds) under diverse streaming scheduling conditions. **Bold**: best; underline: runner-up. Each scenario evaluates schedulers by varying one stress axis (backlog, stream length, load, or capacity) while keeping the others fixed at the default configuration $N_{\text{init}}^{\text{test}}=10$, $N_{\text{stream}}^{\text{test}}=100$, $\Delta^{\text{test}}=25$ s, and $L^{\text{test}}=50$.

| Algorithm | Default | Backlog Robustness | | Statistical Stability | | Load Sensitivity | | Capacity Scaling | |
|---|---|---|---|---|---|---|---|---|---|
| | | $N_{\text{init}}^{\text{test}}=0$ | $N_{\text{init}}^{\text{test}}=30$ | $N_{\text{stream}}^{\text{test}}=50$ | $N_{\text{stream}}^{\text{test}}=200$ | $\Delta^{\text{test}}=12.5s$ | $\Delta^{\text{test}}=50s$ | $L^{\text{test}}=25$ | $L^{\text{test}}=75$ |
| **FIFO** | $310.4_{\pm134.5}$ | $201.2_{\pm138.4}$ | $684.1_{\pm288.1}$ | $277.1_{\pm191.9}$ | $388.5_{\pm284.1}$ | $827.6_{\pm222.1}$ | $76.7_{\pm51.9}$ | $931.6_{\pm404.9}$ | $195.8_{\pm162.9}$ |
| **SJF-CP** | $84.2_{\pm38.7}$ | $69.3_{\pm27.9}$ | $166.7_{\pm72.8}$ | $86.4_{\pm45.8}$ | $79.4_{\pm28.9}$ | $230.3_{\pm93.7}$ | $\mathbf{46.3_{\pm14.9}}$ | $278.9_{\pm121.1}$ | $52.8_{\pm23.1}$ |
| **WFS** | $148.3_{\pm111.2}$ | $107.1_{\pm68.1}$ | $401.2_{\pm298.2}$ | $170.5_{\pm132.0}$ | $120.5_{\pm54.4}$ | $622.2_{\pm383.4}$ | $69.6_{\pm37.6}$ | $1120.8_{\pm796.8}$ | $74.8_{\pm41.7}$ |
| **HEFT** | $129.2_{\pm71.2}$ | $110.4_{\pm53.9}$ | $332.7_{\pm130.2}$ | $141.8_{\pm88.3}$ | $125.1_{\pm45.3}$ | $567.2_{\pm184.6}$ | $58.4_{\pm22.4}$ | $695.6_{\pm362.5}$ | $78.6_{\pm34.0}$ |
| **DRF** | $448.3_{\pm378.3}$ | $434.2_{\pm254.6}$ | $515.8_{\pm387.9}$ | $355.4_{\pm122.7}$ | $215.2_{\pm194.1}$ | $434.2_{\pm120.1}$ | $175.4_{\pm87.5}$ | $516.4_{\pm86.5}$ | $472.4_{\pm147.8}$ |
| **SRPT** | $143.1_{\pm84.9}$ | $113.7_{\pm51.5}$ | $284.8_{\pm99.6}$ | $145.3_{\pm77.1}$ | $156.6_{\pm99.5}$ | $375.3_{\pm109.9}$ | $61.5_{\pm23.4}$ | $412.9_{\pm164.8}$ | $80.7_{\pm36.0}$ |
| **Decima** | $\underline{63.1_{\pm22.2}}$ | $\underline{56.2_{\pm14.7}}$ | $\mathbf{97.1_{\pm37.3}}$ | $\underline{67.4_{\pm25.5}}$ | $\underline{58.7_{\pm11.1}}$ | $\underline{119.1_{\pm46.9}}$ | $51.6_{\pm12.7}$ | $\underline{185.2_{\pm90.2}}$ | $46.9_{\pm12.6}$ |
| **MACE** | $\mathbf{58.6_{\pm14.7}}$ | $\mathbf{53.1_{\pm11.0}}$ | $\underline{100.6_{\pm32.9}}$ | $\mathbf{64.7_{\pm21.0}}$ | $\mathbf{55.2_{\pm8.4}}$ | $\mathbf{118.3_{\pm43.9}}$ | $\underline{49.0_{\pm10.3}}$ | $\mathbf{183.9_{\pm73.5}}$ | $\mathbf{42.5_{\pm8.0}}$ |

Table 2: Average scheduling latency and standard deviation (in milliseconds) per decision under the default configuration $N_{\text{init}}^{\text{test}} = 10$, $N_{\text{stream}}^{\text{test}} = 100$, $\Delta^{\text{test}} = 25$s, $L^{\text{test}} = 50$.

| Metric | FIFO | SJF-CP | WFS | HEFT | DRF | SRPT | Decima | MACE |
|---|---|---|---|---|---|---|---|---|
| **Avg. Latency (ms)** | 0.025 | 2.602 | 0.670 | 7.734 | 4.268 | 1.285 | 5.470 | 5.529 |
| **Std. Deviation (ms)** | 0.131 | 6.682 | 3.086 | 17.181 | 12.993 | 5.108 | 7.435 | 7.844 |

cold-start congestion; (ii) *statistical stability* by changing stream length $N_{\text{stream}}^{\text{test}} \in \{50, 200\}$ to evaluate consistency under varying workload durations; (iii) *load sensitivity* by adjusting inter-arrival times $\Delta^{\text{test}} \in \{12.5\,\text{s}, 50\,\text{s}\}$ for different load intensities; and (iv) *capacity scaling* by modifying available executors $L^{\text{test}} \in \{25, 75\}$ to test adaptability under varying resource level.

MACE achieves the lowest average JCT in 7 out of 9 test settings and outperforms state-of-the-art baselines by up to 9.38%, demonstrating robustness across diverse workload conditions. Its advantage over Decima arises from combining awareness of execution-critical structures with memory-based temporal context, enabling effective prioritization of critical stages and adaptive parallelism control. SJF-CP performs competitively under heavy load, underscoring the importance of critical-path information in guiding scheduling decisions. Static schedulers such as FIFO and WFS degrade substantially under stress due to their inability for dynamic workloads and fluctuating resources.

**Scheduling Latency Analysis** To assess practical responsiveness, we measure scheduling latency per decision under the default configuration. Scheduling latency is the wall-clock time to choose an action once runnable stages become available, including rule evaluation for heuristics and neural inference for learning-based methods. Table 2 reports the average latency over all decisions, and Figure 3 shows how it varies with the number of active stages. (i) Heuristic schedulers such as FIFO, WFS, and SRPT incur negligible overhead. SJF-CP is still lightweight at about 2.6 ms, while HEFT and DRF are more expensive due to ranking and fairness computations. (ii) Learning-based schedulers add a modest cost, with Decima and MACE both around 5-6 ms per decision and exhibiting comparable latency curves as concurrency grows. Notably, their curves terminate at much smaller numbers of active stages than those of the heuristics, indicating that they keep the system less backlogged through more compact execution. Even at the highest concurrency, decision latency stays on the order of a few milliseconds, whereas stage runtimes and

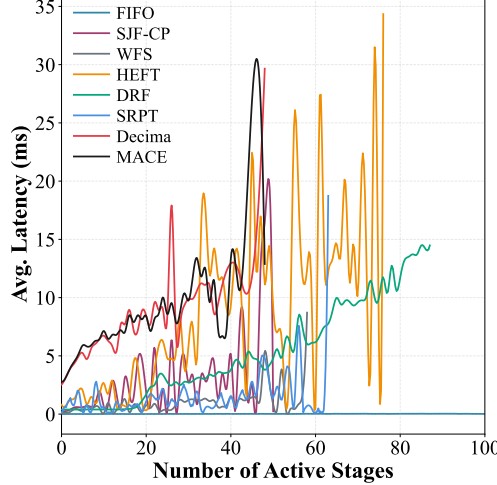

Figure 3: Average scheduling latency (in milliseconds) per decision versus the number of active stages under the default configuration. Each curve is obtained by grouping decisions by the current number of active stages and averaging the latency across test episodes.

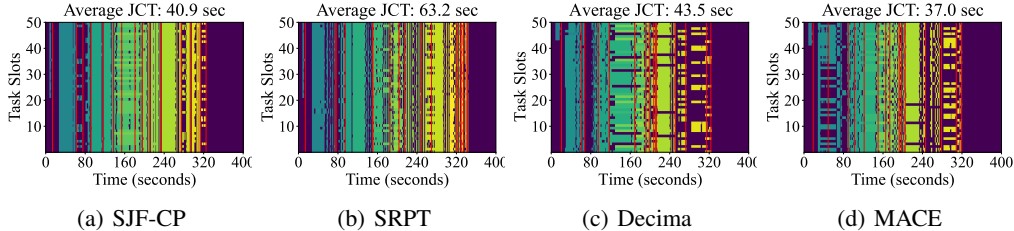

Figure 4: Scheduling traces for a streaming workload. Each subfigure illustrates task execution over time across 50 task slots, where each slot corresponds to an executor running one task at a time. Different DAGs are colored differently. Red vertical lines indicate job completion times, and purple regions denote idle slots.

Table 3: Average job completion time (in seconds) under the default configuration when models are trained on all templates or on subsets with 2/4/6/8 templates held out. All models are evaluated on the full set of templates, measuring generalization to unseen DAG templates.

| Algorithm | All Templates | 2 Held-Out | 4 Held-Out | 6 Held-Out | 8 Held-Out |
|---|---|---|---|---|---|
| **Decima** | 60.1 | 63.1 | 65.6 | 65.5 | 68.9 |
| **MACE** | **49.7** | 51.1 | 51.7 | 53.9 | 53.5 |

inter-arrival intervals are typically on the order of seconds. Thus, this millisecond-level overhead for learning-based methods is negligible at the system timescale and does not become a scheduling bottleneck, a conclusion also reached in the literature (Mao et al., 2019).

**Streaming Trace Analysis**    To understand scheduler behavior under streaming workloads, we visualize the execution traces of 15 DAGs submitted via a Poisson arrival process with a mean inter-arrival time of 25 seconds. Figure 4 compares MACE with three competitive baselines: SJF-CP, SRPT, and Decima. MACE exhibits densely packed execution and early job completions, reflecting efficient task slot usage and the lowest average JCT. SRPT, which prioritizes short jobs based on remaining processing time, often defers long jobs and causes resource underutilization. SJF-CP performs competitively by accelerating critical-path tasks, but its lack of global temporal awareness results in scattered execution and delayed completions. Decima, while improving upon heuristics, still shows inefficiencies with tasks piling up in later stages and earlier slots remaining idle.

**Template-Level Generalization**    Beyond the workload distribution shifts in Table 1, which already test robustness under varying backlog, arrival rate, and cluster capacity, Table 3 further examines template-level generalization under the default configuration. Since heuristic schedulers are rule-based and do not rely on training data, we report only the learned methods MACE and Decima. For each column, the models are trained either on all templates or on subsets where 2/4/6/8 templates are held out, and then evaluated on the full template set. As more templates are excluded from training, the JCT of both methods increases, but MACE increases much more slowly than Decima, e.g., +3.8s vs. +8.8s when holding out 8 templates. This may be because the critical-path-aware encoding and extended receptive field help MACE capture structural motifs that recur across different queries, which in turn leads to stronger generalization to unseen DAG templates.

**Job-Level Scheduling Dynamics**    Figure 5 compares MACE and Decima, examining job-level scheduling dynamics. In subplot (a), we increase job pressure by extending the stream length from 100 to 750, keeping other settings unchanged. MACE maintains fewer concurrent jobs, indicating faster queue clearance and higher throughput. Subplot (b) plots job duration versus total work (i.e., the sum of task runtimes per job). MACE forms a compact cluster near the lower-left corner, indicating that jobs complete faster with less variance. Subplot (c) shows the number of executors allocated per job versus its total work. MACE concentrates more points in the upper-left, meaning that jobs receive more executors. These patterns show that MACE reveals responsiveness by allocating parallelism more effectively, whereas Decima's scattered allocations result in slower completions and higher variance.

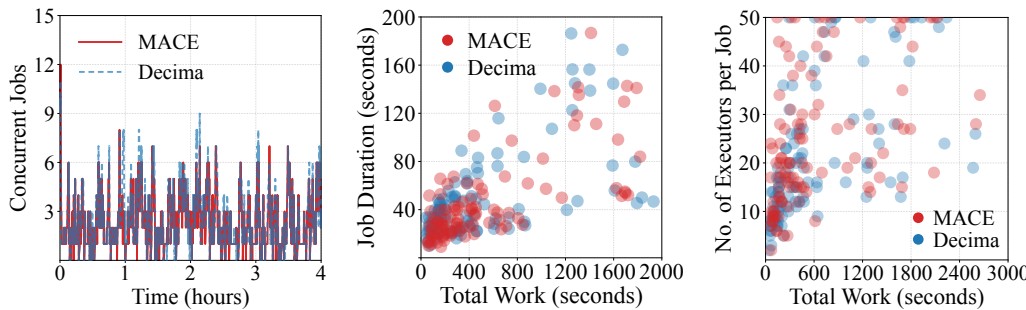

Figure 5: Job-level scheduling behavior of MACE and Decima. (a) Concurrent jobs over time. (b) Job duration versus total work. (c) Executors per job versus total work. MACE completes jobs faster with lower concurrency and more efficient executor use, adapting better to dynamic workloads.

Table 4: Ablation studies of MACE, reporting average JCT (in seconds). **Bold**: best. Each ablated variant removes one key component: memory, positional encoding, or long-range neighbors.

| Scenario | Full MACE | w/o Memory | w/o CP-aware PE | w/o Extended RF |
|---|---|---|---|---|
| **Default** | **49.7** | 55.6 | 58.2 | 53.8 |
| **High Load** | **76.6** | 89.0 | 94.2 | 82.0 |
| **Limited Capacity** | **120.2** | 167.9 | 197.8 | 137.6 |

## 5.2 ABLATION STUDIES

To assess the contributions of MACE's core components, we conduct an ablation study under three representative scenarios: the default configuration, a high-load setting with $\Delta^{\text{test}} = 12.5$s, and a limited-capacity setting with $L^{\text{test}} = 25$. We compare the full model against three variants, each disabling one design: the memory mechanism, the Critical-Path-aware Positional Encoding (CP-aware PE), or the extended Receptive Field (RF). As shown in Table 4, removing CP-aware PE causes the greatest performance drop, highlighting its central role in identifying bottleneck stages. Disabling memory also substantially hurts performance, as it limits the model's use of historical scheduling context. Fixing the receptive field to aggregate only from immediate children moderately increases JCT, indicating the benefit of flexible structural information in adaptive decision making. These results underscore the complementary roles of memory, critical-path encoding, and receptive field control in effective scheduling. We further investigate the impact of memory dimension $d_\theta$ and receptive field size $k$ in Appendix C, showing the sensitivity of MACE to these hyperparameters.

## 6 CONCLUSION

In this study, we present MACE, a memory-augmented reinforcement learning model with critical structure encoding for streaming scheduling. By designing the CSformer encoder and the memory-augmented scheduler, MACE jointly captures execution-critical structures and incorporates historical scheduling contexts, enabling informed and adaptive scheduling decisions. Extensive experiments show that MACE significantly outperforms strong baselines on Spark workloads. For information about the use of large language models during paper preparation, please refer to Appendix D.

**Limitations and Broader Impact** Despite the encouraging results, MACE does not yet account for heterogeneous resources or inter-executor communication costs, which pose more complexities in real-world cluster environments. Our future work will address these limitations and extend the framework to broader and more practical scheduling scenarios. This research may inspire the AI4System community to pay greater attention to these challenges and promote continued advancement in effective streaming scheduling.

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

# A  FEASIBILITY CONSTRAINTS AND PARALLELISM ASSIGNMENT

At decision step $t$, the scheduler selects a runnable stage $v \in \mathcal{S}_t$ and a job-level parallelism $\ell \in \mathcal{L} = \{1, \dots, L\}$ for its job $\mathcal{G}_{j(v)}$, where $j(v)$ indexes the job containing $v$. Let $C_{j(v)}^t$, $F^t$, and $R^t(v)$ denote the executors held by $\mathcal{G}_{j(v)}$, the free executors, and the remaining schedulable tasks on $v$.

**Admissible Parallelism Set**  For $\mathcal{G}_{j(v)}$ at step $t$, the admissible parallelisms are

$$\mathcal{L}_{\text{valid}}(\mathcal{G}_{j(v)}, t) = \{\ell \in \mathcal{L} : \ell \geq C_{j(v)}^t - F^t + 1\}. \tag{19}$$

The threshold $C_{j(v)}^t - F^t + 1$ is the minimal parallelism that ensures a strictly positive increase for the acted job given the current pool of free executors. The parallelism is sampled with a masked softmax over this set:

$$P(\ell \mid \mathbf{o}_{j(v)}^t) = \frac{\exp(\delta_{j(v)}^t(\ell))}{\sum_{\ell' \in \mathcal{L}_{\text{valid}}(\mathcal{G}_{j(v)}, t)} \exp(\delta_{j(v)}^t(\ell'))}. \tag{20}$$

Invalid parallelisms are assigned logits of $-\infty$ and therefore receive zero probability.

**Capacity-Aware Projection**  Given the sampled parallelism $\hat{\ell}$ for $\mathcal{G}_{j(v)}$, define the job-level increase allowed by the chosen parallelism and the current free pool as

$$A_{j(v)}^t = \hat{\ell} - C_{j(v)}^t + F^t. \tag{21}$$

The executed allocation to stage $v$ at step $t$ is

$$\min\{R^t(v), A_{j(v)}^t, F^t\}. \tag{22}$$

This projection, together with the masked sampling of admissible parallelisms, ensures each decision respects stage, job, and per-step capacity constraints.

# B  ADDITIONAL EXPERIMENTAL DETAILS

**Baselines**  FIFO (First-In First-Out) schedules jobs strictly in the order of submission. SJF-CP (Shortest Job First with a Critical-Path proxy) ranks jobs by estimated execution time, calculated from the longest path in the DAG using per-stage durations (Kwok & Ahmad, 1999). WFS (naive Weighted Fair Scheduling) allocates executors proportionally without lookahead or priority adjustments. HEFT (Heterogeneous Earliest Finish Time) selects tasks based on upward rank and assigns them to minimize earliest finish time using insertion-based placement (Topcuoglu et al., 2002). DRF (Dominant Resource Fairness) balances dominant resource shares and reduces to fair sharing in single-resource environments (Ghodsi et al., 2011). SRPT (Shortest Remaining Processing Time) serves the job with the smallest remaining workload (Harchol-Balter, 2013). Decima is a graph-based scheduler trained with reinforcement learning to minimize average JCT (Mao et al., 2019). Heuristic baselines (FIFO/SJF-CP/WFS/HEFT/DRF/SRPT) are implemented following their standard definitions, share the same empirical task-duration statistics as our method, and have no trainable components. Decima follows the official implementation [4] and training recipe provided by the authors, initialized as described in the original paper. Its hyperparameters are carefully tuned on held-out validation seeds to select the best checkpoint, and evaluation is conducted in inference-only mode. For fair comparison, we apply episode parity by generating identical episodes for each evaluation seed, ensuring that all methods face the same job arrivals, DAG identities, and resource settings, with only the scheduling policy varying.

**Implementation**  We train MACE using REINFORCE with the Adam optimizer. The learning rate is selected from $\{1 \times 10^{-4}, 5 \times 10^{-4}, 1 \times 10^{-3}, 2 \times 10^{-3}\}$, with gradient clipping set to $1.0$ and entropy regularization to $10^{-3}$. CSformer is configured with depth ranging from 2 to 6 layers, neighborhood radius $k \in \{1, 2, 3, 4, 5\}$, attention heads $H \in \{2, 4, 8\}$. The memory dimension $d_\theta$ is selected from $\{8, 16, 32, 64, 128\}$, and includes a learned query projection with a multiplicative update gate. Training uses a fixed batch size of $64$, and episodes terminate stochastically. Hyperparameters are tuned via grid search on held-out validation seeds to select the best checkpoint for final evaluation. The environment where we run experiments is:

---

[4](MIT license) https://github.com/hongzimao/decima-sim

- Operating system: Ubuntu 20.04.6 LTS
- CPU information: Intel(R) Xeon(R) Platinum 8378A CPU @ 3.00GHz
- GPU information: NVIDIA A100-SXM4-40GB

## C  HYPERPARAMETER SENSITIVITY ANALYSIS

We evaluate the sensitivity of MACE to two key hyperparameters: the memory dimension $d_\theta$ and the receptive field size $k$. All experiments use the default evaluation setup with $N_{\text{init}}^{\text{test}} = 10$, $N_{\text{stream}}^{\text{test}} = 100$, $\Delta^{\text{test}} = 25\,\text{s}$, and $L^{\text{test}} = 50$.

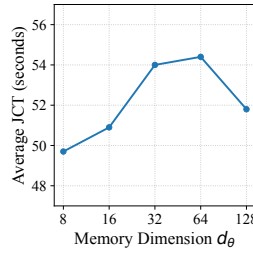 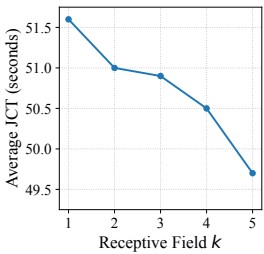

(a) Memory Dimension     (b) Receptive Field

Figure 6: Hyperparameter sensitivity of MACE with varying (a) memory dimension $d_\theta$ and (b) receptive field size $k$ under the default setting.

**Effect of Memory Dimension** We vary $d_\theta$ from 8 to 128 and report the resulting average JCT. As shown in Figure 6 (a), performance degrades as $d_\theta$ increases beyond 16, indicating that a moderate memory size is sufficient. Larger dimensions do not yield further improvements and may introduce redundancy.

**Effect of Receptive Field** We vary the receptive field size $k$ from 1 to 5. As shown in Figure 6 (b), the average JCT decreases consistently as $k$ increases, with the best performance achieved at $k = 5$. This clear downward trend highlights the effectiveness of incorporating long-range structural context when scheduling streaming jobs.

## D  USE OF LARGE LANGUAGE MODELS

In the preparation of this paper, we utilized ChatGPT-4o[5], a large language model developed by OpenAI, to assist in polishing the writing. The model was particularly helpful in refining sentence structure, enhancing clarity, and improving the overall flow of the text. We would like to express our gratitude to the researchers dedicated to both the development and research of large language models, whose work has significantly advanced tools that support and enhance the research process.

---

[5]https://openai.com/chatgpt

