# OpenReview forum: "Toward Neural Streaming Scheduling: A Memory-Augmented Reinforcement Learning Model with Critical Structure Encoding"
_ICLR.cc/2026/Conference — Submitted to ICLR 2026_

### Official Review · Reviewer_PpAN · 2025-10-22

**Soundness:** 2
**Presentation:** 2
**Contribution:** 2
**Rating:** 2
**Confidence:** 4

**Summary:**

This paper introduces MACE, a new reinforcement learning model designed to optimize streaming Directed Acyclic Graph (DAG) scheduling in systems like Spark by minimizing average job completion time. MACE's architecture features two core components: a CSformer encoder that uses "critical-path-aware positional encodings" to identify latency-sensitive stages, and a memory-augmented scheduler that queries a job memory to incorporate historical scheduling decisions and outcomes. Evaluated on the TPC-H benchmark, MACE demonstrated its effectiveness by outperforming state-of-the-art baselines by up to 9.38% across diverse workloads.

**Strengths:**

* The paper clearly identifies two significant weaknesses in existing learning-based schedulers: the failure to explicitly model execution-critical structures like the critical path, and the inability to leverage historical scheduling contexts. It proposes MACE, a new architecture specifically designed to address both of these shortcomings.

* The core components of MACE are well-motivated and their contributions are validated through ablation studies. The CSformer's use of critical-path-aware positional encodings directly targets the critical bottleneck issue, and removing this component causes the largest performance degradation. Similarly, the memory-augmented scheduler provides a concrete mechanism to learn from past decisions, and its removal also substantially hurts performance.

**Weaknesses:**

* The paper's novelty is limited, as it does not introduce a new learning paradigm. It frames DAG scheduling as a reinforcement learning problem, a well-established approach from prior work like Decima. The core contributions are the CSformer and the memory module, which are essentially incremental architectural additions (graph positional encodings and a memory-augmented agent) rather than fundamental advances in either RL or scheduling.

* A critical metric for a "streaming" scheduler, the decision-making latency, is relegated to the appendix. The main paper (pages 1-9) focuses exclusively on job completion time (JCT). Given that the MACE model is demonstrably more complex than its baselines, this overhead is a crucial part of the performance trade-off and its omission from the main experimental results is not acceptable. Please note that the reviewers are not required to read the appendix.

* The empirical evaluation is not well-aligned with the ICLR audience. The experiments are conducted exclusively on Spark using the TPC-H benchmark, which is a traditional data analytics/database workload. The paper fails to evaluate its scheduler on any workloads more central to the modern AI community, such as scheduling for large-scale model training, complex training/data-processing pipelines, or large-batch inference.

* The comparison against other learning-based schedulers is insufficient, as Decima is the only learning-based baseline included in the evaluation. While Decima is a seminal work in this area, the authors explicitly exclude other relevant and more recent learning-based schedulers, such as LACHESIS and DeepWeave. Justifying this exclusion by claiming "incompatible problem formulation"  is a weak defense and prevents a meaningful comparison of MACE against the current state-of-the-art in learning-based DAG scheduling.

* The performance gains are modest and may not justify the added complexity. The paper claims "up to 9.38%" improvement, but in the main "Default" scenario in Table 1, the improvement over the primary learning baseline (Decima) is smaller (58.6 vs 63.1). This marginal gain is achieved by introducing a significantly more complex architecture involving a custom Transformer, a memory matrix, and complex read/write operations.

**Questions:**

* The paper introduces significant architectural complexity (CSformer, a query-based memory matrix) for what appears to be a modest improvement over Decima in the default case (58.6s vs 63.1s). How do you justify this complexity-performance trade-off, especially given that the scheduling latency is a critical metric?

* Speaking of latency, a crucial metric for any "streaming" scheduler is the per-decision overhead. Why was this analysis not included in the main results (e.g., in Table 1) but instead moved to the appendix?

* The evaluation is performed exclusively on the TPC-H benchmark, a traditional data analytics/database workload. Given the ICLR audience, why was the MACE scheduler not evaluated on workloads more representative of modern AI systems, such as scheduling complex ML training pipelines or data-flow graphs for large model inference?

---

> ### Author Response · Authors · 2025-11-20
>
> - We thank the Reviewer for carefully reading our paper and for the thoughtful feedback. We truly appreciate the time and effort devoted to evaluating our work. We have taken all comments into account, updated the manuscript accordingly, and provide point-by-point responses to each question and concern below.
>
> 1. > The paper's novelty is limited, as it does not introduce a new learning paradigm. It frames DAG scheduling as a reinforcement learning problem, a well-established approach from prior work like Decima. The core contributions are the CSformer and the memory module, which are essentially incremental architectural additions (graph positional encodings and a memory-augmented agent) rather than fundamental advances in either RL or scheduling.
>
> **Answer:**
> We appreciate the Reviewer’s concern and would like to clarify our contributions in more detail. Our goal is to advance **how RL is applied** to streaming scheduling, and tackle two **domain-specific challenges**:
> (i) **Execution-critical structures**, modeled via *critical-path-aware encoding and an extended receptive field* defined by shortest-path distances, so the policy can focus on latency-sensitive dependencies;
> (ii) **Historical scheduling contexts**, captured by a *memory-augmented scheduler* that leverages past decisions and rewards to inform current scheduling actions.
> Together, these designs enhance the expressiveness of the RL policy and enable reasoning over both DAG structure and temporal dynamics.
>
> Empirically, Table 1 reports *up to 9.38% improvement over baselines* across diverse workloads, Figure 4 shows *denser execution and faster job completions*, and ablation studies confirm that *each component contributes to performance*. These results indicate that our contributions provide a substantive methodological improvement for RL-based streaming scheduling.
>
> 2. > Speaking of latency, a crucial metric for any "streaming" scheduler is the per-decision overhead. Why was this analysis not included in the main results (e.g., in Table 1) but instead moved to the appendix? How do you justify this complexity-performance trade-off, especially given that the scheduling latency is a critical metric?
>
> **Answer:**
> Thank you for raising this point about per-decision overhead and the complexity-performance trade-off. Since decision latency is on the order of a few **milliseconds**, while stage runtimes and inter-arrival intervals are on the order of **seconds**, this millisecond-level overhead for learning-based methods is acceptable at the system timescale, a conclusion also reached in the literature [1]. For space reasons, the original submission therefore reported detailed per-decision latency in the appendix and only summarized in the main text. In the revised version, we **move the latency table into the main results** and make this overhead–performance trade-off explicit.
>
> We also **add a new experiment (Fig. 3)** that *plots average scheduling latency versus the number of active stages, complementing the aggregate results in Table 2*. From Table 2 and Fig. 3 we observe that: (i) Heuristic schedulers FIFO, WFS, and SRPT incur negligible overhead. SJF-CP remains lightweight at about 2.6 ms, while HEFT and DRF are more expensive due to ranking and fairness computations. (ii) Learning-based schedulers add a modest cost, with MACE and Decima both around 5–6 ms per decision and showing comparable latency curves as concurrency grows. Their curves also terminate at much smaller numbers of active stages than the heuristics, indicating that they keep the system less backlogged through more compact execution. Even at the highest observed concurrency, decision latency stays on the **millisecond scale**, whereas stage runtimes and inter-arrival intervals are typically **second-level**, so the overhead of learning-based methods is acceptable in practice.
>
> We appreciate this question, which prompted us to add the latency-scaling experiment and to present the overhead-benefit trade-off more clearly in the revised manuscript.

---

> > ### Author Response · Authors · 2025-11-20
> > **Official Comment by Authors**
> >
> > 3. > The evaluation is performed exclusively on the TPC-H benchmark, a traditional data analytics/database workload. Given the ICLR audience, why was the MACE scheduler not evaluated on workloads more representative of modern AI systems, such as scheduling complex ML training pipelines or data-flow graphs for large model inference?
> >
> > **Answer:**
> > Thank you for this insightful comment on our evaluation setting and workload choice. Our work focuses on the problem of **streaming scheduling** in data analytics clusters: long-running jobs arrive over time as DAGs and run on a shared pool of identical executors. In this setting, a *streaming TPC-H workload is a natural and widely used benchmark*: each query induces a nontrivial DAG with diverse depth, width, and bottlenecks, and sampling templates, data scales, and Poisson arrivals yields long job streams with substantial concurrency and queueing.
> >
> > We agree that complex ML training pipelines and large-model inference graphs are relevant to the ICLR community, but they typically rely on a **different systems and resource model**, e.g., *GPU/TPU clusters with hardware heterogeneity, explicit communication topologies, CPU-GPU data movement, and data/model/pipeline parallelism*. Scheduling there involves dimensions such as device placement and communication–computation overlap, whereas our study focuses on stage-level scheduling and parallelism on a homogeneous executor pool. Extending MACE to these ML workloads under their corresponding system assumptions is an interesting and complementary direction for future work.
> >
> > 4. > The comparison against other learning-based schedulers is insufficient, as Decima is the only learning-based baseline included in the evaluation. While Decima is a seminal work in this area, the authors explicitly exclude other relevant and more recent learning-based schedulers, such as LACHESIS and DeepWeave. Justifying this exclusion by claiming "incompatible problem formulation" is a weak defense and prevents a meaningful comparison of MACE against the current state-of-the-art in learning-based DAG scheduling.
> >
> > **Answer:**
> > We apologize if our original wording was brief. These methods assume **different system models** (*heterogeneous compute or network-level modeling vs. homogeneous compute*) and operate at **different granularities** (*task placement or coflow-level vs. stage-level scheduling and parallelism*), and are therefore not comparable. Specifically, our setting is a homogeneous streaming cluster where jobs arrive continuously, share a pool of identical executors, and the scheduler repeatedly chooses which stage to run and how much parallelism to allocate. This formulation is a standard scheduling problem in distributed data-processing systems [2,3]. In contrast, LACHESIS addresses DAG scheduling in heterogeneous clusters with task placement and duplication decisions, and DeepWeave focuses on coflow/flow-level scheduling in datacenter networks, making bandwidth and priority decisions at the network layer.
> >
> > In the revised manuscript, we will clarify this with a clearer statement that “learning-based schedulers such as LACHESIS and DeepWeave target heterogeneous task placement or coflow-level network scheduling under different system assumptions and objectives, so we treat them as complementary related work rather than direct baselines for our streaming cluster setting”. Within our setting, *we compare MACE against Decima, to the best of our knowledge, the only learning-based scheduler, as well as a set of production-style heuristic schedulers*. Together, these baselines cover the main established approaches for this scenario.
> >
> > [1] Mao H, Schwarzkopf M, Venkatakrishnan S B, et al. Learning scheduling algorithms for data processing clusters[C]. SIGCOMM 2019.
> > [2] Zaharia M, Chowdhury M, Franklin M J, et al. Spark: Cluster computing with working sets[C]. HotCloud 2010.
> > [3] Carbone P, Katsifodimos A, Ewen S, et al. Apache flink: Stream and batch processing in a single engine[J]. The Bulletin of the Technical Committee on Data Engineering, 2015, 38(4).

---

> ### Author Response · Authors · 2025-11-21
> **Appreciation for Your Reconsideration and Feedback**
>
> We highly appreciate the time and attention you devoted to our rebuttal, and we are genuinely grateful for your willingness to reconsider and adjust the score based on our response. Your review and thoughtful comments have been very helpful in refining both the technical content and the presentation of our work.
>
> Your support has been truly invaluable throughout this iterative process.

---

### Official Review · Reviewer_za5G · 2025-10-27

**Soundness:** 2
**Presentation:** 2
**Contribution:** 2
**Rating:** 4
**Confidence:** 2

**Summary:**

This paper proposes MACE, a novel reinforcement learning (RL) model designed to schedule streaming Directed Acyclic Graph (DAG) jobs in a cluster environment. The primary objective is to learn a scheduling policy that minimizes the average Job Completion Time (JCT). The authors identify two main limitations in prior work: (1) a failure to explicitly model execution-critical structures like critical paths, and (2) an inability to leverage historical scheduling contexts and outcomes.

MACE addresses these gaps with two key architectural components:
1.  **CSformer:** A graph encoder that builds hierarchical embeddings (stage-job-global). It uniquely incorporates **critical-path-aware positional encodings (CP-aware PE)** to prioritize latency-sensitive stages and uses a **tunable attention field** to capture long-range dependencies.
2.  **Memory-Augmented Scheduler:** This component selects a runnable stage and its parallelism. It uses the embeddings from CSformer and queries a **job memory** that stores historical scheduling decisions and rewards, allowing the policy to make more context-aware decisions.

The model is trained end-to-end with policy gradients. Experiments conducted in a Spark simulation using the TPC-H benchmark demonstrate that MACE outperforms several heuristic and learning-based baselines, including Decima, by up to 9.38% on average JCT under diverse workloads.

**Strengths:**

1.  **Well-Motivated Problem and Approach:** The paper tackles a practical and challenging problem in large-scale systems: streaming DAG scheduling. The two motivations (capturing critical structures and using historical context) are clear, intuitive, and directly address significant weaknesses in existing learning-based schedulers.
2.  **Novel Architectural Components:** The core ideas of MACE are technically sound and novel for this domain.
    * The **CSformer**'s use of critical-path-aware positional encodings is a clever way to inject crucial domain knowledge directly into the graph representation, guiding the model's attention toward bottleneck stages. The ablation study strongly supports this, showing that removing the CP-aware PE causes the most significant performance degradation.
    * The **memory-augmented scheduler** provides a principled mechanism to address the short-sightedness of typical RL agents by integrating past experience (decisions and outcomes) into the current decision-making process. This is also well-supported by the ablation study.
3.  **Comprehensive Experimental Evaluation:** The experimental setup is thorough and convincing.
    * The authors compare MACE against a strong set of seven baselines, including both widely-used heuristics (like HEFT, SRPT) and a state-of-the-art learning-based method (Decima).
    * The "stress tests" in Table 1, which vary backlog, stream length, load, and cluster capacity, provide a robust evaluation of MACE's performance and generalization under diverse conditions.
    * The ablation studies (Table 2) are effective in demonstrating the individual contributions of the memory, CP-aware PE, and receptive field, confirming that the components are complementary.
    * The qualitative trace analysis (Figure 3) and job-level dynamics (Figure 4) provide valuable insights into *why* MACE's policies are more efficient, showing denser resource packing and faster queue clearance.

**Weaknesses:**

1.  **Clarity of the Memory Mechanism:** The formulation of the job memory is somewhat confusing. Equation 11 defines the memory $M_{t-1}$ as a transformation $f_m$ over the *entire* set of previous decision trajectories. This implies a potentially unbounded and computationally expensive construction at each step. However, Equations 16 and 17 describe an *update* mechanism ($M_t = M_{t-1} \odot C + U$), which is more typical of a recurrent state. It is unclear how these two formulations reconcile. Is the memory a fixed-size buffer, a growing list, or a recurrent hidden state? This lack of clarity makes it difficult to assess the true scalability and implementation of the memory module.
2.  **Misleading Terminology:** The paper repeatedly refers to a "**tunable** attention field" or "tunable Receptive Field (RF)". However, the description of the method and the hyperparameter analysis (Appendix D, Figure 5(b)) suggest that the field size $k$ is a fixed hyperparameter selected via grid search, not a parameter that is "tuned" dynamically by the model itself. This terminology is misleading. "Configurable" or "extended" receptive field would be more accurate.
3.  **Limited Scalability Analysis:** While Table 3 shows that MACE's scheduling latency per decision (5.529 ms) is acceptable and comparable to Decima's (5.470 ms) for the default setting, this analysis is limited. The paper does not investigate how this latency scales as the cluster state grows—for example, with a much larger number of concurrent jobs (e.g., $N_{init}^{test} \gg 30$) or with significantly more complex DAGs (many more nodes/stages). Both the CSformer's attention mechanism and the memory retrieval step could become bottlenecks in larger, more complex scenarios.
4.  **Baseline Exclusions:** The paper explicitly excludes comparisons to other learning-based schedulers like LACHESIS and DeepWeave, stating their problem formulations

**Questions:**

1) Could you please clarify the exact mechanism of the job memory? How is the memory M_t−1 from Equation 11 constructed and maintained, and how does it relate to the update rule in Equation 16? What is the computational complexity of the memory read (Eq. 13 ) and write (Eq. 16 ) operations, and how do they scale with the episode length?

2) Regarding the "tunable receptive field": Is the radius k a fixed hyperparameter for the entire model, or is it learned or otherwise adapted during execution? If it is a fixed hyperparameter, please consider revising the terminology to "configurable" or "extended" to avoid confusion.

3) The reward function in Equation 18, is an interesting proxy for minimizing average JCT. While cited, could you provide a brief intuition or justification (e.g., via Little's Law) for why minimizing the cumulative sum of this reward (the integral of the number of jobs over time) is equivalent to minimizing the average JCT?


4) Can you comment on the expected scalability of MACE's decision latency (as reported in Table 3 ) with respect to the number of concurrent jobs and the average number of stages per DAG? At what point might the CSformer or memory-augmented scheduler become a practical bottleneck?

---

> ### Author Response · Authors · 2025-11-20
> **Rebuttal by Authors**
>
> - We sincerely thank the Reviewer for the careful reading of our paper and for the constructive comments. We truly appreciate the time and effort invested in evaluating our work. We have carefully considered all comments, revised the manuscript accordingly, and provide point-by-point responses to each question and concern below.
>
> 1. > (i) Could you please clarify the exact mechanism of the job memory? (ii) How is the memory M_t−1 from Equation 11 constructed and maintained, and how does it relate to the update rule in Equation 16?  (iii) What is the computational complexity of the memory read (Eq. 13 ) and write (Eq. 16 ) operations, and how do they scale with the episode length?
>
> **Answer:**
> We apologize for the confusion about the memory formulation.
> (i) The job memory is a **fixed-size matrix** shared by all jobs and maintained as a **recurrent hidden state** throughout an episode. At each decision step $t$, the scheduler first reads from memory $\mathbf{M}\_{t-1}$: for each job $i$, we construct a query $\mathbf{o}\_i^t$, and compute
> $\mathbf{m}\_i^t = \mathbf{M}\_{t-1}\cdot q(\mathbf{o}\_i^t)$,
> yielding a job-specific summary fed into the parallelism head.
> After selecting the next stage, we form a context vector $\hat{\mathbf{o}}\_t$ for the associated job, and update the memory online:
> $\mathbf{M}\_t = \mathbf{M}\_{t-1} \odot \mathbf{C}\_\theta(\hat{\mathbf{o}}^t) + \mathbf{U}\_\phi(\hat{\mathbf{o}}^t)$,
> where $\mathbf{C}\_\theta(\hat{\mathbf{o}}^t), \mathbf{U}\_\phi(\hat{\mathbf{o}}^t) \in \mathbb{R}^{d_m \times d_m}$ are produced by small feedforward networks and act as a retention gate and an additive update, respectively.
> (ii) Eq. (11) in the original draft was meant as a semantic description that $\mathbf{M}\_{t-1}$ summarizes the past trajectory up to time $t-1$. The specific recurrent update rule is defined in Eq. (16). *The revised version removes Eq. (11), explicitly states in text that the job memory is a fixed-size recurrent state shared across jobs, and keeps Eq. (16) as the concrete recursive definition of $\mathbf{M}\_t$*.
> (iii) The memory read computing $\mathbf{m}\_i^t = \mathbf{M}\_{t-1}\cdot q(\mathbf{o}\_i^t)$ is one matrix–vector product over a $d_m \times d_m$ matrix and a $d_m$-dimensional vector, i.e., $O(d_m^2)$. Considering we query memory for $J^t$ jobs, the read cost at that step is $O(J^t d_m^2)$. The memory write computing $\mathbf{C}\_\theta(\hat{\mathbf{o}}^t)$ and $\mathbf{U}\_\phi(\hat{\mathbf{o}}^t)$ via small MLPs and updating
> $\mathbf{M}\_t = \mathbf{M}\_{t-1} \odot \mathbf{C}\_\theta(\hat{\mathbf{o}}^t) + \mathbf{U}\_\phi(\hat{\mathbf{o}}^t)$
> over $d_m \times d_m$ matrices is also $O(d_m^2)$. Thus, for an episode with $T$ decision steps, *the total time complexity of the memory module is $O(TJ^td_m^2)$ and the space complexity is $O(d_m^2)$*.
>
> We thank the Reviewer for these questions, which helped us refine both the notation and the description of the memory mechanism in the revised manuscript.
>
> 2. > Regarding the "tunable receptive field": Is the radius k a fixed hyperparameter for the entire model, or is it learned or otherwise adapted during execution? If it is a fixed hyperparameter, please consider revising the terminology to "configurable" or "extended" to avoid confusion.
>
> **Answer:**
> We thank the Reviewer for pointing out this potential source of confusion. The radius $k$ is not learned or adapted online. It is a **fixed structural hyperparameter** selected on a validation set. Our use of "tunable receptive field" was intended to indicate that CSformer’s structural receptive field can be tuned via $k$ to trade off between local context and longer-range dependencies, and our hyperparameter sensitivity analysis (Appendix C) shows that varying $k$ affects performance.
>
> In the revised version, we replace "tunable receptive field" with the more precise term **extended receptive field**. We appreciate the Reviewer’s comment, which helped us clarify this point.

---

> > ### Author Response · Authors · 2025-11-20
> > **Rebuttal by Authors**
> >
> > 3. > The reward function in Equation 18, is an interesting proxy for minimizing average JCT. While cited, could you provide a brief intuition or justification (e.g., via Little's Law) for why minimizing the cumulative sum of this reward (the integral of the number of jobs over time) is equivalent to minimizing the average JCT?
> >
> > **Answer:**
> > We are happy to clarify the intuition behind the reward in Eq. (18). On each interval $[T^{t-1}, T^{t})$, let $J^t$ be the number of jobs in the system and
> > $\Delta T^t = T^{t} - T^{t-1}$. The per-step reward is
> > $r^t = - \Delta T^t J^t$,
> > so the cumulative reward over an episode is
> > $\sum_t r^t = -\sum_t \Delta T^t J^t \approx -\int N(u)\,du$,
> > where $N(u)$ is the number of jobs in the system at time $u$. Each job $i$ arriving at time $A_i$ and completing at time $C_i$ contributes $1$ to $N(u)$ for all $u \in [A_i, C_i)$, so its contribution to the area under $N(u)$ is $C_i - A_i$. Summing over all jobs,
> > $\int N(u)\,du = \sum_i (C_i - A_i)$,
> > which is the total JCT of all jobs. For an episode with $|\mathcal{J}|$ jobs, where *$|\mathcal{J}|$ is fixed for a given arrival stream, minimizing the total JCT is equivalent to minimizing average JCT*.
> >
> > 4. > Can you comment on the expected scalability of MACE's decision latency (as reported in Table 3 ) with respect to the number of concurrent jobs and the average number of stages per DAG? At what point might the CSformer or memory-augmented scheduler become a practical bottleneck?
> >
> > **Answer:**
> > This is an insightful question about the scalability and practical impact of MACE’s decision latency. To make this clearer, we **added Fig. 3**, which *plots average scheduling latency versus the number of active stages*, complementing the aggregate results in Table 2. These results show that: (i) for all methods except FIFO, latency increases roughly linearly with concurrency, with some fluctuations. (ii) Learning-based schedulers add a **modest cost**, with Decima and MACE both around 5-6 ms per decision and exhibiting comparable latency curves as concurrency grows. Notably, their curves terminate at much smaller numbers of active stages than those of the heuristics, indicating that they keep the system less backlogged through more compact execution. Even at the highest concurrency, decision latency stays on the order of a few **milliseconds**, whereas stage runtimes and inter-arrival intervals are typically on the order of **seconds**. Thus, *this millisecond-level overhead for learning-based methods is acceptable at the system timescale, a conclusion also reached in the literature* [1].
> >
> > MACE would only risk becoming a bottleneck in much more extreme regimes, such as very large clusters with many thousands of simultaneously active stages and workloads dominated by very short stages. In such cases, standard engineering techniques such as reducing model depth or width, pruning the candidate stage set, caching and incrementally updating graph embeddings, or using batched and hardware-accelerated inference can further reduce latency.
> >
> > We thank the Reviewer for raising this question, which motivated us to add the latency-scaling experiment and to clarify the empirical behavior of MACE’s decision latency.

---

> > > ### Author Response · Authors · 2025-11-20
> > > **Rebuttal by Authors**
> > >
> > > 5. > The paper explicitly excludes comparisons to other learning-based schedulers like LACHESIS and DeepWeave, stating their problem formulations.
> > >
> > > **Answer:**
> > > We apologize if our original wording was brief. These methods assume **different system models** (*heterogeneous compute or network-level modeling vs. homogeneous compute*) and operate at **different granularities** (*task placement or coflow-level vs. stage-level scheduling and parallelism*), and are therefore not comparable. Specifically, our setting is a homogeneous streaming cluster where jobs arrive continuously, share a pool of identical executors, and the scheduler repeatedly chooses which stage to run and how much parallelism to allocate. This formulation is a standard scheduling problem in distributed data-processing systems [2,3]. In contrast, LACHESIS addresses DAG scheduling in heterogeneous clusters with task placement and duplication decisions, and DeepWeave focuses on coflow/flow-level scheduling in datacenter networks, making bandwidth and priority decisions at the network layer.
> > >
> > > In the revised manuscript, we will clarify this with a clearer statement that “learning-based schedulers such as LACHESIS and DeepWeave target heterogeneous task placement or coflow-level network scheduling under different system assumptions and objectives, so we treat them as complementary related work rather than direct baselines for our streaming cluster setting”. Within our setting, *we compare MACE against Decima, to the best of our knowledge, the only learning-based scheduler, as well as a set of production-style heuristic schedulers*. Together, these baselines cover the main established approaches for this scenario.
> > >
> > > [1] Mao H, Schwarzkopf M, Venkatakrishnan S B, et al. Learning scheduling algorithms for data processing clusters[C]. SIGCOMM 2019.
> > > [2] Zaharia M, Chowdhury M, Franklin M J, et al. Spark: Cluster computing with working sets[C]. HotCloud 2010.
> > > [3] Carbone P, Katsifodimos A, Ewen S, et al. Apache flink: Stream and batch processing in a single engine[J]. The Bulletin of the Technical Committee on Data Engineering, 2015, 38(4).

---

> > > > ### Comment · Reviewer_za5G · 2025-11-26
> > > >
> > > > Thank you for the addressed clarification and weaknesses

---

> > > > > ### Author Response · Authors · 2025-11-27
> > > > > **Appreciation for Your Follow-Up and Clarifications**
> > > > >
> > > > > We sincerely thank you for carefully reading and responding to our rebuttal!
> > > > >
> > > > > Your questions about (i) the memory mechanism, (ii) the extended receptive field, and (iii) the scalability of decision latency have greatly helped us refine both our presentation and experiments in the revised manuscript.
> > > > >
> > > > > If you feel that these additional explanations and experiments address your main concerns, we would be very grateful if you could consider reflecting this in your final scores.
> > > > >
> > > > > **In any case, we truly appreciate the time and care you have devoted to our work**.

---

### Official Review · Reviewer_4nDf · 2025-10-30

**Soundness:** 2
**Presentation:** 3
**Contribution:** 2
**Rating:** 4
**Confidence:** 4

**Summary:**

The paper proposes an RL scheduler for streaming DAG jobs, built on the Decima framework but extending the action space to jointly choose the next runnable stage and its parallelism. It encodes whole-DAG structure with a critical-path–aware transformer and conditions decisions on a compact memory of past scheduling outcomes. The policy is trained end-to-end to minimise average job completion time (JCT) and is evaluated through simulation.

**Strengths:**

1. Dealing with critical paths in DAG jobs and long historical patterns are well motivated in the scheduling context;

2. Encoding the scheduling stage as a position in the critical path of the DAG is novel.

**Weaknesses:**

1. The evaluation is done using simulation only compared to Decima.

2. As there are only 22 DAG templates used in evaluation and the memory augmentation mechanism remembers many combinations, the generalisability of the scheduling policy can be limited to different DAGs.

3. Even though the critical path encoding is a new approach for characterising the job DAG, it is questionable whether critical paths need to be explicitly modelled in scheduling as the GNN graph summary in Decima already implicitly captures the representation of a DAG through GNN. Critical paths can be used as a post-hoc feature to derive interpretable scheduling policies, but the graph summary itself might be more effective in training a optimised scheduling policy.

**Questions:**

The results in Fig.3 shows SJF outperforms Decima. This is contrary to the batch job results in the Decima paper. Any explanation to the results?  How is the Decima model trained in the evaluation?

---

> ### Author Response · Authors · 2025-11-20
> **Rebuttal by Authors**
>
> - We sincerely thank the Reviewer for the careful reading of our paper and the constructive feedback. We appreciate the time and effort invested in evaluating our work. We have carefully taken these comments into account, revised the manuscript accordingly, and provide detailed point-by-point responses below.
>
> 1. > The evaluation is done using simulation only compared to Decima.
>
> **Answer:**
> We appreciate the Reviewer’s concern regarding the evaluation setup. Our work targets **streaming scheduling**, a practical and challenging problem in distributed data-processing systems [1,2] yet relatively underexplored in the AI community. Within this setting, *Decima, to the best of our knowledge, is the only learning-based baseline. Following Decima, we also compare against common production-style heuristic schedulers*. Table 1 and Figure 4 report results for MACE against Decima and these heuristics under a range of workload and cluster conditions.
>
> In the line of work on scheduling for data analytics clusters, *evaluating new schedulers in a Spark-based simulator has become standard practice [3,4,5]*, as it captures job DAG structures, execution-time characteristics, and executor constraints while allowing controlled variation of workloads and system parameters. Following this methodology and prior work [5], we conduct our evaluation in a Spark simulation environment, where we generate diverse TPC-H–based job streams (via different queries, data scales, and Poisson arrivals) and systematically stress-test along multiple axes (initial backlog, arrival rate, stream length, and cluster size). This setup allows a comprehensive comparison of MACE, Decima, and heuristic baselines under varied load and resource conditions.
>
> 2. > As there are only 22 DAG templates used in evaluation and the memory augmentation mechanism remembers many combinations, the generalisability of the scheduling policy can be limited to different DAGs.
>
> **Answer:**
> We thank the Reviewer for raising this concern about generalization beyond the 22 TPC-H templates.
> (i) The 22 query templates induce a **highly diverse** workload for training and evaluation. Each streaming workload is sampled from a **large combinatorial space** over *(template, data scale, random seed, initial backlog, arrival rate, cluster size, stream length)*, yielding many distinct DAG instances and job streams with different stage runtimes, critical paths, and concurrency patterns. Thus, in the evaluation results of Table 1, the policy encounters job combinations that were never seen during training, rather than replaying a small fixed library.
> (ii) To directly probe **template-level generalization**, we **add Table 3** in the revised version. Under the default configuration, we *train MACE and Decima either on all templates or on subsets where 2/4/6/8 templates are held out*, and then always evaluate on the full template set. As more templates are excluded from training, JCT increases for both methods, but *MACE increases much more slowly* (e.g., +3.4s vs.\ +8.8s when holding out 8 templates). This is likely because the critical-path-aware encoding and extended receptive field allow MACE to capture structural motifs that recur across different queries, leading to stronger generalization to unseen DAG templates.
> (iii) The memory module does not memorize templates. Reads and writes are keyed by job embeddings together with past parallelism choices and rewards, therefore learns **abstract scheduling patterns** for structurally similar jobs, rather than per-template lookup rules.
>
> We appreciate this comment, which prompted us to clarify the template-level generalization setting and add targeted experiments in the revised manuscript.

---

> > ### Author Response · Authors · 2025-11-20
> > **Rebuttal by Authors**
> >
> > 3. > Even though the critical path encoding is a new approach for characterising the job DAG, it is questionable whether critical paths need to be explicitly modelled in scheduling as the GNN graph summary in Decima already implicitly captures the representation of a DAG through GNN. Critical paths can be used as a post-hoc feature to derive interpretable scheduling policies, but the graph summary itself might be more effective in training a optimised scheduling policy.
> >
> > **Answer:**
> > We thank the Reviewer for raising this question, which goes to the core of whether critical paths should be modeled explicitly or left for the GNN to learn implicitly. GNNs **struggle to capture long critical paths**, *limited to 1-WL expressive power [6] and prone to over-squashing [7*]. At the same time, RL optimizes long-term JCT with high-variance rewards, making it *hard to infer criticality from noisy returns*, and *global pooling tends to blur which specific stages are critical and how deep they are*.
> >
> > Our critical-path-aware encoding is therefore introduced as an explicit **inductive bias** rather than a post-hoc feature. For each stage we encode (i) its depth along the critical path and (ii) whether it lies on the critical path, and inject these signals via positional encodings into the per-stage embedding consumed by CSformer. This *exposes critical-path structure during representation learning*, and the policy *learns to trade it off against the current cluster state and other execution-critical structures*, deciding when to prioritize critical-path stages and when to deviate.
> >
> > Empirically, the ablation "w/o CP-aware PE" in Table 4 shows the largest degradation in average JCT among our variants. Moreover, the SJF-CP heuristic, which also explicitly uses critical-path length, consistently outperforms CP-agnostic heuristics. Together, these results indicate that explicitly encoding critical-path structure provides substantial benefit.
> >
> > 4. > The results in Fig.3 shows SJF outperforms Decima. This is contrary to the batch job results in the Decima paper. (i) Any explanation to the results? (ii) How is the Decima model trained in the evaluation?
> >
> > **Answer:**
> > Thank you for the careful reading and for pointing out this discrepancy.
> > (i) Fig. 3 is not intended to establish the overall ranking between SJF-CP and Decima, but to **visualize execution behaviour** on a single streaming episode (15 DAGs) for interpretability. On this particular sample, SJF-CP happens to achieve a slightly lower average JCT than Decima. *Our conclusions about relative performance are based on the aggregated results in Table 1*, which report average JCT over many random seeds and episodes under nine test configurations (varying backlog, stream length, arrival interval, and cluster size). There, *Decima outperforms SJF-CP in most settings*, while SJF-CP remains a strong and often competitive heuristic. The difference from the original Decima paper stems from the problem setting: their results focus on batch scheduling with a fixed set of known jobs, whereas our work targets streaming scheduling with Poisson arrivals on a continuously running cluster.
> > (ii) As detailed in Appendix B, we use the authors’ public implementation of Decima in the same Spark-based simulator and TPC-H–based streaming workload as for MACE, tune its hyperparameters on validation seeds. During evaluation, MACE, Decima, and all heuristic baselines are run on exactly the same randomly generated job streams, ensuring a fair comparison across all methods.
> >
> > [1] Zaharia M, Chowdhury M, Franklin M J, et al. Spark: Cluster computing with working sets[C]. HotCloud 2010.
> > [2] Carbone P, Katsifodimos A, Ewen S, et al. Apache flink: Stream and batch processing in a single engine[J]. The Bulletin of the Technical Committee on Data Engineering, 2015, 38(4).
> > [3] Venkataraman S, Yang Z, Franklin M, et al. Ernest: Efficient performance prediction for Large-Scale advanced analytics[C]. NSDI 2016.
> > [4] Grandl R, Ananthanarayanan G, Kandula S, et al. Multi-resource packing for cluster schedulers[J]. ACM SIGCOMM Computer Communication Review, 2014, 44(4): 455-466.
> > [5] Mao H, Schwarzkopf M, Venkatakrishnan S B, et al. Learning scheduling algorithms for data processing clusters[C]. SIGCOMM 2019.
> > [6] Xu K, Hu W, Leskovec J, et al. How powerful are graph neural networks?[C]. ICLR 2019.
> > [7] Alon U, Yahav E. On the bottleneck of graph neural networks and its practical implications[C]. ICLR 2021.

---

### Official Review · Reviewer_riU2 · 2025-11-01

**Soundness:** 2
**Presentation:** 2
**Contribution:** 2
**Rating:** 4
**Confidence:** 3

**Summary:**

This paper addresses streaming DAG scheduling and proposes **MACE**, which consists of two main modules:
- The **CSformer** encoder enriches node embeddings on the critical path (CP) by injecting CP-related information (Equations 2 and 4). The rest of the structure—concatenating stage, job, and global embeddings—is similar to  Mao et al. [1], except that the authors adopt a Transformer-based architecture instead of GNNs.
- The **Memory-Augmented Scheduler** performs stage selection in the same way as [1], while in parallelism selection, the authors use historical decision information (previous job embeddings, parallelism choices, and rewards) as additional embeddings to enhance the current state.

Overall, the work focuses on *input feature and embedding-level design* while relying on a standard REINFORCE algorithm for training. The introduction of Transformer-style encoding into scheduling is interesting and extends architectural diversity in this domain, but the dataset and baselines are limited and outdated.

----

[1] Mao et al., *Learning Scheduling Algorithms for Data Processing Clusters*, SIGCOMM 2019.
[2] Jeon et al., *Neural DAG Scheduling via One-Shot Priority Sampling*, ICLR 2023.
[3] Qi et al., *Reinforcement Learning for One-Shot DAG Scheduling with Comparability Identification and Dense Reward*, NeurIPS 2025.

**Strengths:**

* The paper introduces a Transformer-based encoder for scheduling problems.
* The integration of critical-path encoding is meaningful, enabling better structural awareness in scheduling decisions.
* Incorporating memory into the parallelism selection process provides a novel perspective on leveraging historical scheduling decisions.

**Weaknesses:**

* The paper mainly improves feature and embedding design without methodological novelty in the RL formulation or training procedure.
* Baselines are very old — only compared with Decima [1], omitting many recent RL-based and learning-based schedulers from 2021–2025.
* The dataset is restricted to TPC-H, lacking standard benchmarks such as Pegasus or job shop datasets, limiting the generality of the conclusions..
* Related work is outdated, missing key references on DAG/workflow scheduling from recent top-tier conferences.

**Questions:**

1. The terminology throughout the paper (e.g., “stage”) does not align with standard DAG-scheduling terminology (“task”). Please ensure consistent and conventional use of terms.
1. In the *Learning-based Scheduling* subsection of Related Work, most cited papers are from 2019–2022. Please update with recent literature (e.g., ICLR, NeurIPS, ICML, TPDS, TSC 2023–2025) that addresses workflow or streaming scheduling.
1. Could you clarify the role of Equation (3)? How is it used in practice, and what impact does it have on training or inference?
1. The **dataset** should include more established benchmarks such as Pegasus workflows or Job Shop Scheduling (JSP) instances, as used in recent DAG-scheduling papers [2][3].
1. **Baseline** selection is limited to [1]; please compare with more recent RL-based scheduling methods to demonstrate improvements in both performance and inference time.
1. Regarding Equations (11)–(14): since these include historical stage information of a job, does this violate the Markov property required by the MDP formulation? Please clarify how your model maintains or approximates the MDP assumption.

---

> ### Author Response · Authors · 2025-11-20
> **Rebuttal by Authors**
>
> - We sincerely thank the Reviewer for the careful reading of our paper and the constructive feedback. We greatly appreciate the time and effort invested in evaluating our work. We have carefully considered all comments and revised the manuscript accordingly. Below, we provide point-by-point responses to each question and concern.
>
> 1. > (i) In the Learning-based Scheduling subsection of Related Work, most cited papers are from 2019–2022. Please update with recent literature (e.g., ICLR, NeurIPS, ICML, TPDS, TSC 2023–2025) that addresses workflow or streaming scheduling. (ii) Baseline selection is limited to Decima; please compare with more recent RL-based scheduling methods to demonstrate improvements in both performance and inference time. (iii) The dataset should include more established benchmarks such as Pegasus workflows or Job Shop Scheduling (JSP) instances, as used in recent DAG-scheduling papers.
>
> **Answer:**
> We apologize for any confusion about the scope of our setting. Because our setting **differs in system assumptions, inputs, and outputs** from static DAG/workflow/JSP scheduling, the suggested baselines and benchmarks are *not comparable*.
>
>  As stated in the Problem Description section, our paper focuses on the practical and challenging problem of **streaming scheduling**: *DAG jobs arrive as a stochastic stream with unknown future arrivals, and multiple jobs share a pool of identical executors*. At each scheduling event, the scheduler decides *which stage to run next and how much parallelism to allocate per job*. This streaming formulation is a standard scheduling problem in distributed data-processing systems [1,2], yet remains relatively underexplored in the AI community. By contrast, the recent works [3,4], including one-shot DAG/workflow/JSP schedulers, fall under **static scheduling**: *a complete DAG/workflow/JSP instance (or fixed set of instances) is known in advance, and a policy outputs a global ordering or mapping per instance*, without modeling an unbounded job stream or per-job parallelism on a shared executor pool.
>
> This distinction underlies our choices in Related Work, baselines, and benchmarks.
> (i) The “Learning-based Scheduling” subsection is intentionally scoped to methods for this streaming setting. *In the revised manuscript, Related Work now begins with an explicit separation between static and streaming scheduling, and cites recent learning-based methods [3,4] in the static category, while keeping the detailed discussion focused on streaming scheduling.*
> (ii) Under this setting, Decima [5] is, to the best of our knowledge, the only learning-based scheduler, so our experiments compare MACE in depth against Decima and classical schedulers, reporting both performance (Table 1, Fig. 4, Fig. 5) and decision latency (Table 2, Fig. 3).
> (iii) Following prior work [5], our benchmark choice is tied to this streaming setting, whereas Pegasus workflows and JSP instances are standard benchmarks for static workflow/JSP scheduling.
>
> We thank the Reviewer for these comments, which helped us further refine the scope and presentation of the paper.
>
> 2. > The paper mainly improves feature and embedding design without methodological novelty in the RL formulation or training procedure.
>
> **Answer:**
> We appreciate the Reviewer’s concern and would like to clarify our contributions in more detail. Our goal is to advance **how RL is applied** to streaming scheduling, and tackle two **domain-specific challenges**:
> (i) **Execution-critical structures**, modeled via *critical-path-aware encoding and an extended receptive field* defined by shortest-path distances, so the policy can focus on latency-sensitive dependencies;
> (ii) **Historical scheduling contexts**, captured by a *memory-augmented scheduler* that leverages past decisions and rewards to inform current scheduling actions.
> Together, these designs enhance the expressiveness of the RL policy and enable reasoning over both DAG structure and temporal dynamics.
>
> Empirically, Table 1 reports *up to 9.38% improvement over baselines* across diverse workloads, Figure 4 shows *denser execution and faster job completions*, and ablation studies confirm that *each component contributes to performance*. These results indicate that our contributions provide a substantive methodological improvement for RL-based streaming scheduling.

---

> > ### Author Response · Authors · 2025-11-20
> > **Rebuttal by Authors**
> >
> > 3. > The terminology throughout the paper (e.g., “stage”) does not align with standard DAG-scheduling terminology (“task”). Please ensure consistent and conventional use of terms.
> >
> > **Answer:**
> > We thank the Reviewer for carefully reading our paper. Our terminology is chosen to match the standard abstraction in Spark-like clusters [1,2]. A job is decomposed into multiple stages, and **each stage consists of many executor-level tasks** that process data partitions in parallel. Our scheduler operates at the stage level, rather than making decisions for individual tasks.
> >
> > *In the traditional DAG-scheduling literature, the DAG node typically called a “task” corresponds to what Spark calls a stage, while “tasks” in Spark are the parallel instances inside that node.* For readers familiar with Spark-like systems, keeping the job-stage-task hierarchy explicit is therefore more faithful to the underlying system and avoids overloading “task” at two different levels. In the revised version, we will re-check the manuscript to ensure this hierarchy is used consistently.
> >
> > 4. > (i) Could you clarify the role of Equation (3)? (ii) How is it used in practice, and (iii) what impact does it have on training or inference?
> >
> > **Answer:**
> > Thanks for your comments.
> > (i) Eq. (3) defines the **extended receptive field** of CSformer’s structural attention. For each stage $v$, it specifies $N_k^{\text{unexec}}(v) = \lbrace u \in V \mid \operatorname{dist}(v,u) \le k,\ u \text{ is unexecuted} \rbrace$, i.e., the set of *unexecuted* stages that $v$ is allowed to attend to. Because this receptive field is defined via *shortest-path distance* on the job DAG, CSformer can look beyond immediate parents and children and capture *execution-critical structures* emphasized in the Introduction.
> > (ii) Operationally, Eq. (3) is used as the **attention neighborhood** in Eq. (5): both the attention weights $\alpha^{(h)}_{vu}$ and the aggregated representation $\mathbf{z}^{(h)}_v$ are computed only over $u \in N^{\text{unexec}}_k(v)$. The same mechanism is applied during training and inference, and the radius $k$ is a hyperparameter that controls how far each stage can “see’’ along the DAG.
> > (iii) Our ablation study (Table 4) and the sensitivity experiments over $k$ (Appendix C) show the impact of this design: removing the extended receptive field makes MACE lose long-range structural information and leads to noticeably worse average JCT.
> >
> > 5. > Regarding Equations (11)–(14): since these include historical stage information of a job, does this violate the Markov property required by the MDP formulation? Please clarify how your model maintains or approximates the MDP assumption.
> >
> > **Answer:**
> > This is an interesting question about how our memory module interacts with the MDP assumption. Including historical stage information via Equations (11)-(14) **does not violate the MDP assumption**. It simply *enriches the policy’s input* with a learned summary of past decisions and outcomes while the underlying environment remains a Markov decision process.
> >
> > Specifically, given the environment state $s_t$ and the current action $a_t$ (chosen stage and its parallelism), the environment deterministically advances execution, updates the cluster, and computes the reward $r_t$, so
> > $P(s_{t+1}, r_t \mid s_0,\dots,s_t, a_0,\dots,a_t) = P(s_{t+1}, r_t \mid s_t, a_t)$.
> > The environment defines a standard MDP over cluster states. Equations (11)-(14) introduce a *job-level memory* used only by the policy. They compress historical scheduling context into a matrix $M_t$. From an RL perspective, this is equivalent to working with an **augmented state** $\tilde{s}\_t = (s\_t, M\_t)$, where $M_t$ is updated deterministically from $(M_{t-1}, s_{t-1}, a_{t-1}, r_{t-1})$. Given $\tilde{s}\_t$ and $a_t$, *the next augmented state $\tilde{s}\_{t+1} = (s_\{t+1}, M_\{t+1})$ is fully determined by the environment dynamics and the memory update rule*, so the overall process remains Markov.
> >
> > [1] Zaharia M, Chowdhury M, Franklin M J, et al. Spark: Cluster computing with working sets[C]. HotCloud 2010.
> > [2] Carbone P, Katsifodimos A, Ewen S, et al. Apache flink: Stream and batch processing in a single engine[J]. The Bulletin of the Technical Committee on Data Engineering, 2015, 38(4).
> > [3] Jeon W, Gagrani M, Bartan B, et al. Neural DAG scheduling via one-shot priority sampling[C]. ICLR. 2023.
> > [4] Qi X, Zhang D, Liu T, et al. Reinforcement learning for one-shot DAG scheduling with comparability identification and dense reward[C]. NeurIPS 2025.
> > [5] Mao H, Schwarzkopf M, Venkatakrishnan S B, et al. Learning scheduling algorithms for data processing clusters[C]. SIGCOMM 2019.

---

### Author Response · Authors · 2025-11-21
**Summary of Revisions and Clarifications**

Dear Reviewers,

We sincerely appreciate the time and effort you have devoted to evaluating our submission. Your feedback has been invaluable in improving the work.

In this note, we briefly summarize several recurring **misunderstandings** and **main concerns** in the reviews to facilitate a fair assessment.

The most critical **misunderstanding** concerns:
- **Problem setting & baselines:** Some comments appear to treat our **streaming scheduling** as equivalent to static DAG/workflow scheduling, leading to **misplaced expectations** of Pegasus/JSP-style datasets and static schedulers. In the revision, we *more clearly distinguish streaming scheduling from static DAG/workflow/JSP settings*, and explain why certain methods are not comparable.

The main **concerns** focus on:
- **Methodological contribution:** Our goal is not to propose a new RL framework, but to advance **how RL is applied** to streaming scheduling, explicitly addressing two *domain-specific challenges*.
- **Decision latency:** Some comments question when the overhead of learning-based methods might become a bottleneck. We report that measured decision latency is at the **millisecond** level, acceptable compared to **second**-level job execution times and *consistent with prior systems work*, and we **add latency–stage experiments** to further illustrate this.
- **Model generalization:** We clarify that job streams are sampled from a **large combinatorial space**, and we **add template-holdout experiments** to further support our generalization claims.

In the revised manuscript, **all newly added experimental results are highlighted in $\color{red}{\text{red}}$**, and **all revised clarifications are highlighted in $\color{purple}{\text{purple}}$** for ease of reference.

Our problem setting, streaming scheduling in distributed data-processing systems, is a **practical and challenging systems problem** that *remains relatively underexplored in the AI community*. We hope this work helps draw the attention of AI researchers to this direction and is well aligned with the ICLR track on *learning on time series and dynamical systems*.

Best regards,
Authors of the submission

---

### Meta-Review · Area_Chair_bWEU · 2026-01-08

**Summary:**

The author expresses gratitude for the reviewers' feedback, which has been instrumental in refining their work. They address several misunderstandings and criticisms highlighted in the reviews.

**Benefits:**
1. The paper focuses on streaming scheduling in distributed data-processing systems, a practical and underexplored area in the AI community.
2. The authors aim to enhance the application of reinforcement learning (RL) in this context, addressing specific challenges rather than proposing a new RL framework.
3. They provide evidence that decision latency is negligible and consistent with prior systems and have included additional experiments to support this claim.

**Decision:**

The recommendation is to improve the paper before being accepted due to significant misunderstandings among reviewers regarding its problem setting and contributions, particularly the conflation of streaming and static scheduling. Furthermore, questions about the methodological contributions, the generalization capabilities of the proposed model and the dataset evaluation.

**Reviewer Concerns:**

**Main Criticisms:**
1. Misunderstandings regarding the problem setting and baselines, with some reviewers conflating streaming scheduling with static scheduling.
2. Concerns about the placement of latency results, with one reviewer noting that these results are relegated to the appendix.
3. Questions regarding the methodological contribution and generalization of their model, particularly in relation to the overhead of learning-based methods.

**Addressing Concerns:**
The authors clarify that their work is distinct from static scheduling and emphasize the practical implications of their research. They address the placement of latency results by reiterating their acceptability and providing additional experiments to illustrate decision latency. To tackle concerns about model generalization, they explain the sampling of job streams from a large combinatorial space and include template-holdout experiments to reinforce their claims. Overall, the authors aim to redirect attention to the significance of their work within the AI research community.

**Reviewer Scores:**

Honestly, a few reviewers would have changed their score, mainly due to misunderstanding the goal of the paper and its contribution to the community. In my opinion the paper presents an intriguing problem, but it should be tackled as a benchmark to attract the AI community to such a problem. Moreover, the main concern about the dataset is addressed indirectly, saying that we tackle a different but similar problem. My recommendation is to show that the algorithm is able to tackle similar datasets and show how good the problem is, having less information about the DAG.

---

### Decision · Program_Chairs · 2026-01-26

Reject